# Allosteric activation or inhibition of PI3Kγ mediated through conformational changes in the p110γ helical domain

Noah J Harris[1†], Meredith L Jenkins[1†], Sung-Eun Nam[2], Manoj K Rathinaswamy[1], Matthew AH Parson[1], Harish Ranga-Prasad[1], Udit Dalwadi[2], Brandon E Moeller[1], Eleanor Sheeky[1], Scott D Hansen[3], Calvin K Yip[2]*, John E Burke[1,2]*

[1]Department of Biochemistry and Microbiology, University of Victoria, Victoria, Canada; [2]Department of Biochemistry and Molecular Biology, The University of British Columbia, Vancouver, Canada; [3]Department of Chemistry and Biochemistry, Institute of Molecular Biology, University of Oregon, Eugene, United States

*For correspondence:
calvin.yip@ubc.ca (CKY);
jeburke@uvic.ca (JEB)

†These authors contributed
equally to this work

Competing interest: See page
18

Reviewing Editor: Amy H
Andreotti, Iowa State University,
United States

**Abstract** PI3Kγ is a critical immune signaling enzyme activated downstream of diverse cell surface molecules, including Ras, PKCβ activated by the IgE receptor, and Gβγ subunits released from activated GPCRs. PI3Kγ can form two distinct complexes, with the p110γ catalytic subunit binding to either a p101 or p84 regulatory subunit, with these complexes being differentially activated by upstream stimuli. Here, using a combination of cryo electron microscopy, HDX-MS, and biochemical assays, we have identified novel roles of the helical domain of p110γ in regulating lipid kinase activity of distinct PI3Kγ complexes. We defined the molecular basis for how an allosteric inhibitory nanobody potently inhibits kinase activity through rigidifying the helical domain and regulatory motif of the kinase domain. The nanobody did not block either p110γ membrane recruitment or Ras/Gβγ binding, but instead decreased ATP turnover. We also identified that p110γ can be activated by dual PKCβ helical domain phosphorylation leading to partial unfolding of an N-terminal region of the helical domain. PKCβ phosphorylation is selective for p110γ-p84 compared to p110γ-p101, driven by differential dynamics of the helical domain of these different complexes. Nanobody binding prevented PKCβ-mediated phosphorylation. Overall, this work shows an unexpected allosteric regulatory role of the helical domain of p110γ that is distinct between p110γ-p84 and p110γ-p101 and reveals how this can be modulated by either phosphorylation or allosteric inhibitory binding partners. This opens possibilities of future allosteric inhibitor development for therapeutic intervention.

## eLife assessment

This study presents **fundamental** new insight into the regulatory apparatus of PI3Kγ, a kinase in signaling pathways that control the immune response and cancer. A suite of biophysical and biochemical approaches provide **convincing** evidence for new sites of allosteric control over enzyme activity. The rigorous findings provide structure and dynamic information that may be exploited in efforts to control PI3Kγ activity in a therapeutic setting.

## Introduction

The class I phosphoinositide 3 kinases (PI3Ks) are master regulators of myriad functions through their generation of the lipid signaling molecule phosphatidylinositol 3,4,5-trisphosphate ($PIP_3$) downstream of cell surface receptors (*Burke and Williams, 2015*; *Rathinaswamy and Burke, 2020*;

*Vanhaesebroeck et al., 2021*; *Vasan and Cantley, 2022*). The class I PI3Ks can be further subdivided into the class IA and class IB subfamilies, with class IB PI3Ks being critical in immune signaling, and are composed of a single p110γ catalytic subunit that can bind to either a p101 or p84 regulatory subunit (*Hawkins and Stephens, 2015*; *Lanahan et al., 2022*; *Okkenhaug, 2013*). The two PI3Kγ complexes (either p110γ-p84 or p110γ-p101) play essential and independent roles in both the adaptative and innate immune systems. Class IB PI3Kγ has shown promise as a therapeutic target, primarily as an immunomodulator of the tumor microenvironment leading to enhanced anti-tumor immune responses (*De Henau et al., 2016*; *Kaneda et al., 2016b*). Multiple isoform selective small molecule ATP competitive inhibitors of p110γ are in clinical trials for multiple forms of human cancers (*Li et al., 2021*). However, all inhibitors currently developed toward p110γ act as ATP competitive inhibitors, showing equal potency against both p110γ-p84 and p110γ-p101 complexes.

Detailed experiments on the role of p110γ in mice show that knockout of both p101 and p84 leads to PIP$_3$ levels that are equivalent to knockout of p110γ, showing that all cellular PI3Kγ activities require the presence of either a p84 or p101 regulatory subunit (*Rynkiewicz et al., 2020*). The two complexes are differentially activated by membrane localized receptors, including G-protein coupled receptors (*Li et al., 2000*; *Stephens et al., 1997*), Ras (*Jin et al., 2020*; *Kurig et al., 2009*), toll-like receptors (*Luo et al., 2018*), and the IgE antigen receptor (*Laffargue et al., 2002*; *Walser et al., 2013*). This leads to the different complexes driving unique immune responses, with p110γ-p101 involved in chemotaxis in neutrophils (*Bohnacker et al., 2009*; *Deladeriere et al., 2015*), and p110γ-p84 involved in reactive oxide production. Differential activation of unique PI3Kγ complexes downstream of GPCRs and Ras is caused by the ability of p101 to directly bind to Gβγ subunits downstream of activated GPCRs, with this being lost in p84, making p110γ-p84 activation by Gβγ dependent on Ras-mediated membrane recruitment (*Rathinaswamy et al., 2023*; *Kurig et al., 2009*; *Rynkiewicz et al., 2020*). Activation of PI3Kγ downstream of the IgE antigen receptor is driven by calcium-mediated activation of protein kinase C, leading to the selective phosphorylation and activation of p110γ at S582 (*Walser et al., 2013*), with this putatively only occurring in p110γ-p84 and not in p110γ-p101. The full molecular mechanisms underlying how phosphorylation of p110γ is selective for different p84 or p101 complexes, and how it activates lipid kinase activity are poorly understood.

Extensive biophysical approaches including cryo electron microscopy (cryo-EM), X-ray crystallography, and hydrogen-deuterium exchange mass spectrometry (HDX-MS) have provided extensive insight into the molecular underpinnings of how p110γ associates with both p101 and p84, how they are differentially activated by Ras and GPCR signals, and how they can be activated on lipid membranes (*Pacold et al., 2000*; *Walker et al., 1999*; *Rathinaswamy et al., 2021a*; *Rathinaswamy et al., 2021c*; *Gangadhara et al., 2019*; *Vadas et al., 2013*; *Rathinaswamy et al., 2023*). The p110γ catalytic subunit is composed of an adaptor binding domain (ABD), a Ras binding domain (RBD), a C2 domain, a helical domain, and a bi-lobal kinase domain (*Rathinaswamy et al., 2021a*; *Walker et al., 1999*). A set of helices positioned C-terminal to the activation loop in the kinase domain play a critical role in regulating activity, with this region referred to as the regulatory motif (*Rathinaswamy et al., 2021c*). The p110γ isoform is unique in that it is inhibited in the absence of a regulatory subunit, with this driven by an autoinhibitory conformation of the regulatory motif, that is proposed to require membrane association to disrupt (*Gangadhara et al., 2019*). The regulatory motif is a common site of activating mutations in the other class I PI3K isoforms (*Jenkins et al., 2023*), with p110γ having rare activating oncogenic mutants in this region (*Rathinaswamy et al., 2021c*). The p110γ subunit interacts with either p84 or p101 at an interface composed of the C2 domain, and the linkers between the RBD-C2 and C2-helical domains (*Rathinaswamy et al., 2023*). The p110γ-p84 complex forms a more dynamic complex compared to p110γ-p101 (*Rathinaswamy et al., 2023*; *Shymanets et al., 2013*), however, no clear unique regulatory role of this difference in dynamics has been identified.

The fundamental roles of p110γ in inflammatory processes have made it a therapeutic target in many pathological conditions, including asthma (*Campa et al., 2018*), arthritis (*Camps et al., 2005*), obesity (*Becattini et al., 2011*; *Breasson et al., 2017*), and cancer (*De Henau et al., 2016*; *Kaneda et al., 2016b*; *Kaneda et al., 2016a*). There are significant side effects from non-isoform selective PI3Kγ inhibitors (*Bohnacker et al., 2017*; *Vanhaesebroeck et al., 2021*), which has driven the development of highly p110γ selective small molecule inhibitors (*Bell et al., 2012*; *Evans et al., 2016*; *Gangadhara et al., 2019*). However, all p110γ inhibitors will target both p110γ-p101 and p110γ-p84, so there is a potential for the development of allosteric inhibitors outside of the ATP binding site.

Initial promise has been reported for the development of class IA p110α oncogene-specific allosteric inhibitors. However, further investigation of the molecular mechanisms underlying p110γ regulation will be required for the discovery of regions that can be targeted for allosteric inhibitor development.

Here, we report critical roles of the helical domain of p110γ in both activation and inhibition of lipid kinase activity. We characterized an allosteric inhibitory nanobody (NB7) that potently inhibits p110γ activity. Cryo-EM was used to define the inhibitory interface, which is composed of the helical domain, the ABD-RBD linker, and the regulatory motif of the kinase domain of p110γ. The region that the nanobody binds to is in close spatial proximity to a previously identified PKCβ phosphorylation site (S582) in the helical domain, and oncogenic activating mutants in the regulatory motif. We fully characterized the activity and dynamics of stoichiometrically PKCβ phosphorylated p110γ, leading to the discovery of a novel additional phosphorylation site (either S594 or S595). PKCβ phosphorylation was highly selective for p110γ and p110γ-p84, with limited phosphorylation of p110γ-p101. HDX-MS analysis showed that phosphorylation of p110γ leads to unfolding of the N-terminal region of the helical domain, and increased kinase activity. The presence of the inhibitory nanobody significantly blocks PKCβ phosphorylation, while phosphorylation of p110γ prevented binding to NB7. Overall, this work provides unique insight into the critical role of the helical domain in controlling p110γ activity, and how phosphorylation and binding partners can modify this regulation. It also reveals a unique binding site located at the interface of the helical and kinase domain that can be targeted for future allosteric inhibitor design.

## Results

### Molecular mechanism of nanobody inhibition of p110γ

We previously identified multiple nanobodies that inhibited the activity of p110γ. One from this group (denoted NB7 throughout the manuscript) potently inhibited the membrane-mediated activation of p110γ-p84 by both Ras and Gβγ, with HDX-MS experiments mapping the NB7 binding interface to the RBD, helical and kinase domains (*Rathinaswamy et al., 2021b*). We originally hypothesized that NB7 worked by sterically inhibiting Ras binding to the RBD of p110γ. To further explore the molecular mechanism of inhibition we purified all complexes of p110γ (p110γ apo, p110γ-p84, p110γ-p101) along with the NB7 nanobody. The SDS-PAGE of all proteins utilized in this study are shown in *Source data 1*.

To define the mechanism concerning how NB7 inhibits PI3Kγ activity we analyzed how this nanobody inhibited all class IB PI3Kγ complexes (p110γ, p110γ-p84, p110γ-p101) upon activation by lipidated Gβγ subunits. Intriguingly, we found that all three forms of p110γ were potently inhibited by NB7 (*Figure 1A*). While the $IC_{50}$ measured for the three complexes was different, this is likely mainly due to the dramatic difference in protein required to measure lipid kinase activity in vitro (~300 nM for p110γ apo/p110γ-p84, and ~10 nM for p110γ-p101). This suggested that the mechanism of inhibition was not driven by a steric block of Ras association through the RBD, as previously proposed (*Rathinaswamy et al., 2021b*). We examined the binding of this nanobody to all complexes using biolayer interferometry (BLI). The nanobody bound equivalently tightly to all complexes, with ~2 nM potency for p110γ, p110γ-p84, and p110γ-p101 (*Figure 1B*). We also tested binding of the nanobody to all class IA PI3Ks, and there was no detectable binding to p110α, p110β, and p110δ (*Figure 1C*).

To further understand the mechanism by which this nanobody blocked lipid kinase activity we measured the bulk membrane recruitment dynamics of fluorescently labeled Dy647-p84-p110γ on supported lipid bilayers (SLBs) using total internal reflection fluorescence (TIRF) microscopy. We found that the nanobody had no effect on membrane recruitment of p110γ-p84 on bilayers containing membrane-tethered Ras (GTP) and Gβγ (*Figure 1D–F*). Membrane binding was not affected when the nanobody was spiked into samples containing membrane-associated Dy647-p84-p110γ (*Figure 1D*). Similarly, pre-incubation of Dy647-p84-p110γ with 500 nM NB7 did not perturb membrane association of the kinase when flowed over a supported membrane (*Figure 1E*).

We wanted to define the molecular mechanism of how nanobody NB7 was a potent allosteric inhibitor of lipid kinase activity. We purified the complex of nanobody NB7 bound to p110γ-p84 to homogeneity by gel filtration. Using this sample, we obtained a cryo-EM reconstruction of the complex of nanobody (NB7)-bound p110γ at 3.0 Å overall resolution from 149,603 particles (*Figure 2A–D*, *Figure 2—figure supplements 1 and 2* and *Supplementary file 1*). The density map was of sufficient

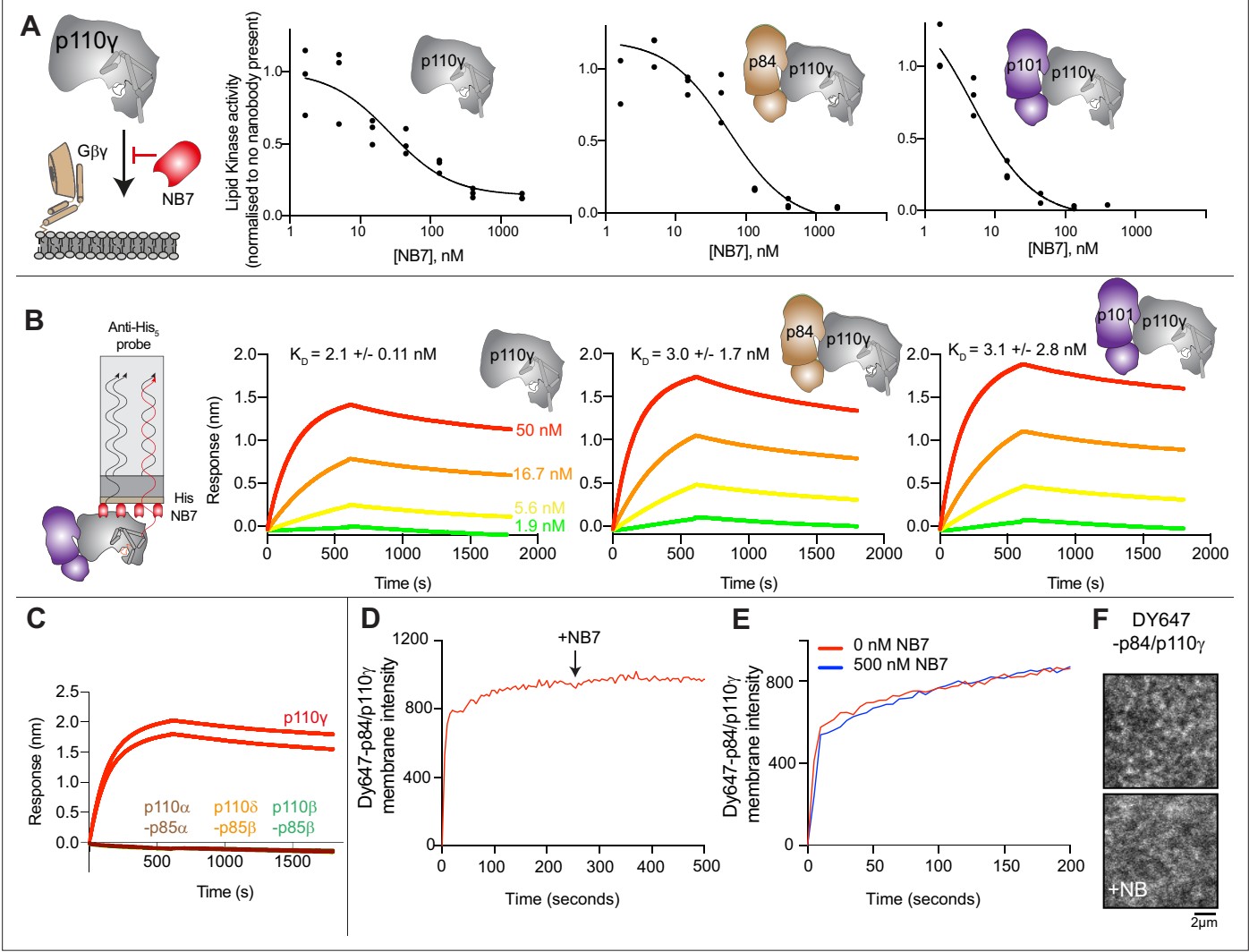

**Figure 1.** The inhibitory nanobody NB7 binds tightly to all p110γ complexes and inhibits kinase activity, but does not prevent membrane binding. (**A**) Cartoon schematic depicting nanobody inhibition of activation by lipidated Gβγ (1.5 μM final concentration) on 5% PIP₂ membrane (5% phosphatidylinositol 4,5-bisphosphate [PIP₂], 95% phosphatidylserine [PS]) activation. Lipid kinase assays showed a potent inhibition of lipid kinase activity with increasing concentrations of NB7 (3–3000 nM) for the different complexes. Experiments are carried out in triplicate (n=3) with each replicate shown. The y-axis shows lipid kinase activity normalized for each complex activated by Gβγ in the absence of nanobody. Concentrations of each protein were selected to give a lipid kinase value in the detectable range of the ATPase Transcreener assay. The protein concentration of p110γ (300 nM), p110γ-p84 (330 nM), and p110γ-p101 (12 nM) was different due to intrinsic differences of each complex to be activated by lipidated Gβγ and is likely mainly dependent for the difference seen in NB7 response. (**B**) Association and dissociation curves for the dose response of His-NB7 binding to p110γ, p110γ-p84, and p110γ-p101 (50–1.9 nM) are shown. A cartoon schematic of biolayer interferometry (BLI) analysis of the binding of immobilized His-NB7 to p110γ is shown on the left. Dissociation constants (K_D) were calculated based on a global fit to a 1:1 model for the top three concentrations and averaged with error shown. Error was calculated from the association and dissociation value (n=3) with standard deviation shown. Full details are present in *Source data 1*. (**C**) Association and dissociation curves for His-NB7 binding to p110γ, p110α-p85, p110β-p85, and p110δ-p85. Experiments were performed in duplicate with a final concentration of 50 nM of each class I phosphoinositide 3 kinase (PI3K) complex. (**D**) Effect of NB7 on PI3Kγ recruitment to supported lipid bilayers containing H-Ras (GTP) and farnesyl-G as measured by total internal reflection fluorescence microscopy (TIRF-M). DY647-p84/p110γ displays rapid equilibration kinetics and is insensitive to the addition of 500 nM nanobody (black arrow, 250 s) on supported lipid bilayers containing H-Ras (GTP) and farnesyl-G. (**E**) Kinetics of 50 nM DY647-p84/p110γ membrane recruitment appears indistinguishable in the absence and presence of nanobody. Prior to sample injection, DY647-p84/p110γ was incubated for 10 min with 500 nM nanobody. (**F**) Representative TIRF-M images showing the localization of 50 nM DY647-p84/p110γ visualized in the absence or presence of 500 nM nanobody (+NB7). Membrane composition for panels C–E: 93% DOPC, 5% DOPS, 2% MCC-PE, Ras (GTP) covalently attached to MCC-PE, and 200 nM farnesyl-G.

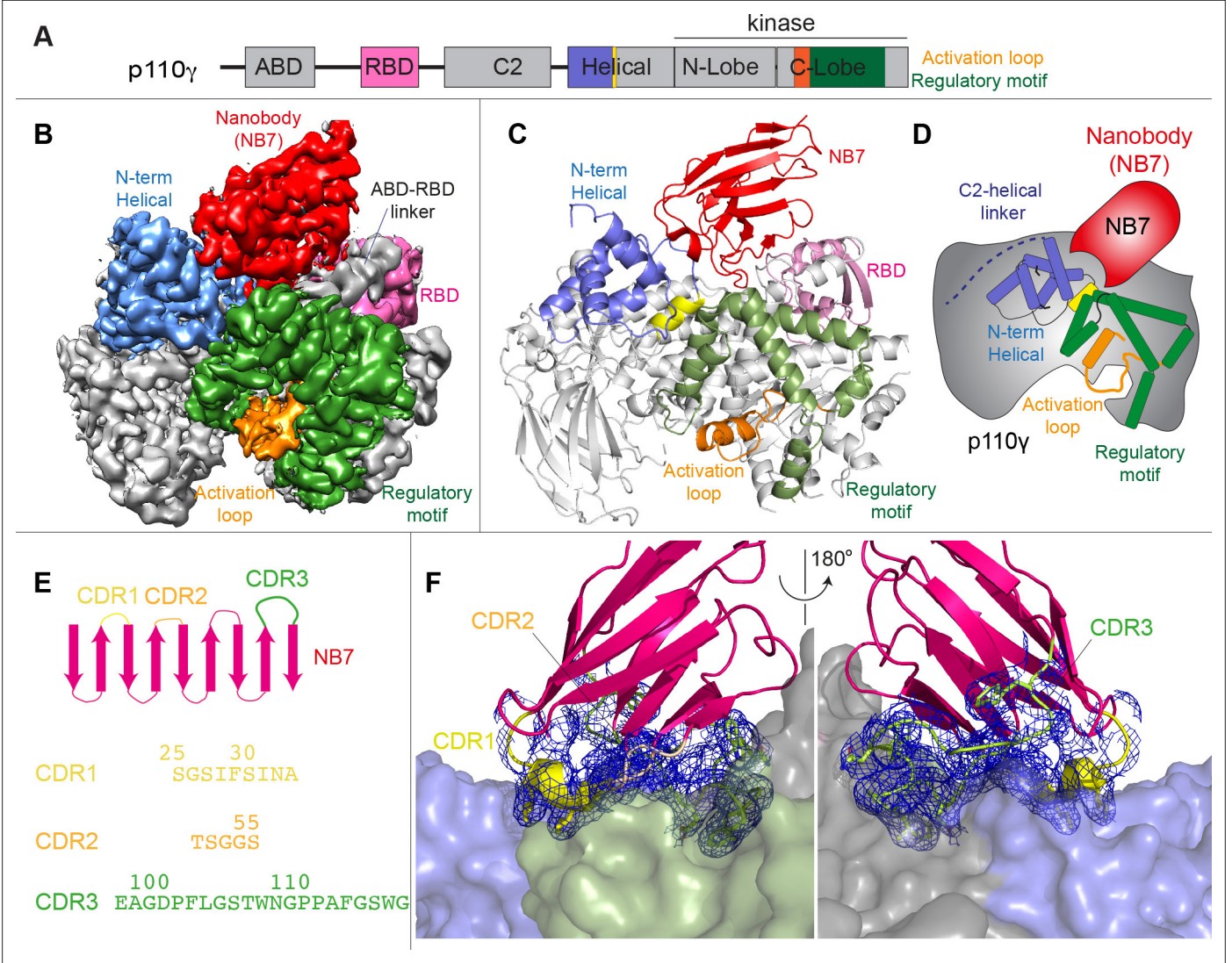

**Figure 2.** Structure of p110γ bound to inhibitory nanobody NB7. (**A**) Domain schematics of p110γ with helical domain (blue), activation loop (orange), and regulatory motif (green) of p110γ annotated. (**B**) Cryo electron microscopy (cryo-EM) density of the p110γ-NB7 complex colored according to the schematic in (**A**). (**C**) Cartoon model of the structure of p110γ bound to NB7 colored according to (**A**). (**D**) Schematic depicting the key features of p110γ and the nanobody binding site, colored according to panel (**A**). (**E**) Domain schematic of NB7 complementarity determining regions (CDRs) and their sequences. (**F**) Zoom in on the binding interface of NB7, with the CDRs colored as in panel E, and the electron density of the CDRs contoured at 3σ (blue mesh).

The online version of this article includes the following figure supplement(s) for figure 2:

**Figure supplement 1.** p110γ-NB7 complex cryo electron microscopy (cryo-EM) analysis workflow.

**Figure supplement 2.** Density fit of p110γ-NB7 complex.

**Figure supplement 3.** Comparison of full-length p110γ bound to NB7 compared to p110γ-p101.

quality to perform automated and manual construction of the p110γ-NB7 complex, with unambiguous building of the interfacial contacts between NB7 and p110γ (*Figure 2—figure supplement 2*). Nanobody binding did not induce any large-scale conformational changes of p110γ, as the structure of p110γ bound to NB7 was similar to the apo p110γ crystal structure or p110γ-p101 cryo-EM structure (*Figure 2—figure supplement 3*). The lowest local resolution was in the ABD, with increased B-factors of the ABD in the p110γ-NB7 structure compared to p110γ-p101 (*Figure 2—figure supplement 3*). This is consistent with the concept that ABD flexibility plays an important role in class I PI3K regulation (*Liu et al., 2022*).

The interface between NB7 and p110γ was extensive, with ~1200 Å of buried surface area, with interactions of the ABD-RBD linker, N-terminus of the helical domain, and the regulatory motif at the turn between k8 and k9 (1022-1026aa). This location in the regulatory motif is where both activating oncogenic (R1021C) and inhibitory loss of function mutants have been identified (R1021P) (*Takeda et al., 2019*), as well as a putative inhibitory phosphorylation site (T1024) (*Perino et al., 2011*). The resolution was sufficient to unambiguously build the three complementarity determining region (CDR) loops of NB7 that mediate target selectivity (*Figure 2E–F*). The interface is primarily hydrophobic, with only 8 hydrogen bonds, and 1 electrostatic interaction among the 33 interfacial residues of NB7. A pocket formed between the helical domain and the ABD-RBD linker forms the majority of the interface, with extensive interactions with the long CDR3. The CDR1 loop packed up against the N-terminal section of the helical domain, with the CDR2 loop forming the interface with the regulatory motif. Previous study of oncogenic mutants in the regulatory motif of p110γ showed that increased dynamics mediated by these mutants increased kinase activity, putatively by breaking the autoinhibitory tryptophan lock in k12 of the regulatory motif (*Rathinaswamy et al., 2021c*). Therefore, rigidifying the regulatory motif likely explains the molecular basis for how it prevents kinase activity. The nanobody interface is distinct from the predicted Gβγ interface (*Rathinaswamy et al., 2023*) and the experimentally resolved Ras interface (*Pacold et al., 2000*), explaining why it can still be membrane recruited by these stimuli.

The interface of NB7 with p110γ is distant from both the putative membrane binding surface and the catalytic machinery of the kinase domain. To further understand how this nanobody could potently inhibit PI3Kγ activity we examined any other potential modulators of PI3Kγ activity localized in this region. There are two regulatory phosphorylation sites in the helical (*Walser et al., 2013*) and kinase domain (*Perino et al., 2011*) localized at the NB7 interface. This is intriguing as helical domain phosphorylation is activating, and kinase domain phosphorylation is inhibitory. This suggested that a critical role in the regulation of p110γ is the dynamics of this kinase-helical interface. To fully define the role of NB7 in altering the dynamics of the helical domain we needed to study other modulators of helical domain dynamics.

## p110γ activation by helical domain phosphorylation

To further understand the potential role of helical domain dynamics in controlling p110γ activity we examined the role of S582 helical domain phosphorylation by the protein kinase PKCβII (encoded by the gene *PRKCB2*, referred to as PKCβ for simplicity throughout the manuscript) (*Walser et al., 2013*). S582 is located on the interior of the helical domain and would not be expected to be exposed when the N-terminal region of the helical domain is folded (*Figure 3A*). To understand this better at a molecular level, we purified a catalytic fragment of PKCβ and performed protein phosphorylation reactions on p110γ apo, p110γ-p84, and p110γ-p101. We identified a phosphorylated peptide containing S582, and surprisingly, we found an additional p110γ phosphorylation site at either S594/S595 (*Figure 3B*, *Figure 3—figure supplement 1*). The modification at this site results in a single phosphorylation event, but due to CID MS/MS fragmentation we cannot determine which site is modified, and we will refer it as S594/S595 throughout the manuscript. The S594/S595 site is also located in the N-terminal region of the helical domain, and is even more buried than S582, and would not be expected to be exposed when this region is folded (*Figure 3A*). Dose-response curves of PKCβ treatment were carried out for p110γ (*Figure 3C*), p110γ-p84 (*Figure 3D*), and p110γ-p101 (*Figure 3E*). Both p110γ and p110γ-p84 showed similar dose-response curves for PKCβ treatment, with similar curves for S582 and S594/S595. The p110γ-p101 complex was only very weakly phosphorylated, with <100-fold lower levels compared to p110γ and p110γ-p84 (*Figure 3E*). This is consistent with the helical domain in p110γ being more rigid when bound to p101, compared to either bound to p84 or p110γ alone.

To provide additional insight into the molecular mechanisms underlying p110γ phosphorylation we carried out HDX-MS experiments on p110γ and phosphorylated p110γ (90.8% phosphorylated S594/595, 92% phosphorylated S582) (*Figure 4A*). The full data underlying the experiment is available in *Source data 1*, and data processing information is in *Supplementary file 2*. We have previously observed that the N-terminal region of the helical domain of apo p110γ (residues spanning 557-630aa) shows isotope profiles that are consistent with EX1 hydrogen-deutruim (H/D) exchange kinetics (*Rathinaswamy et al., 2021b*; *Rathinaswamy et al., 2021c*; *Vadas et al., 2013*; *Walser et al., 2013*).

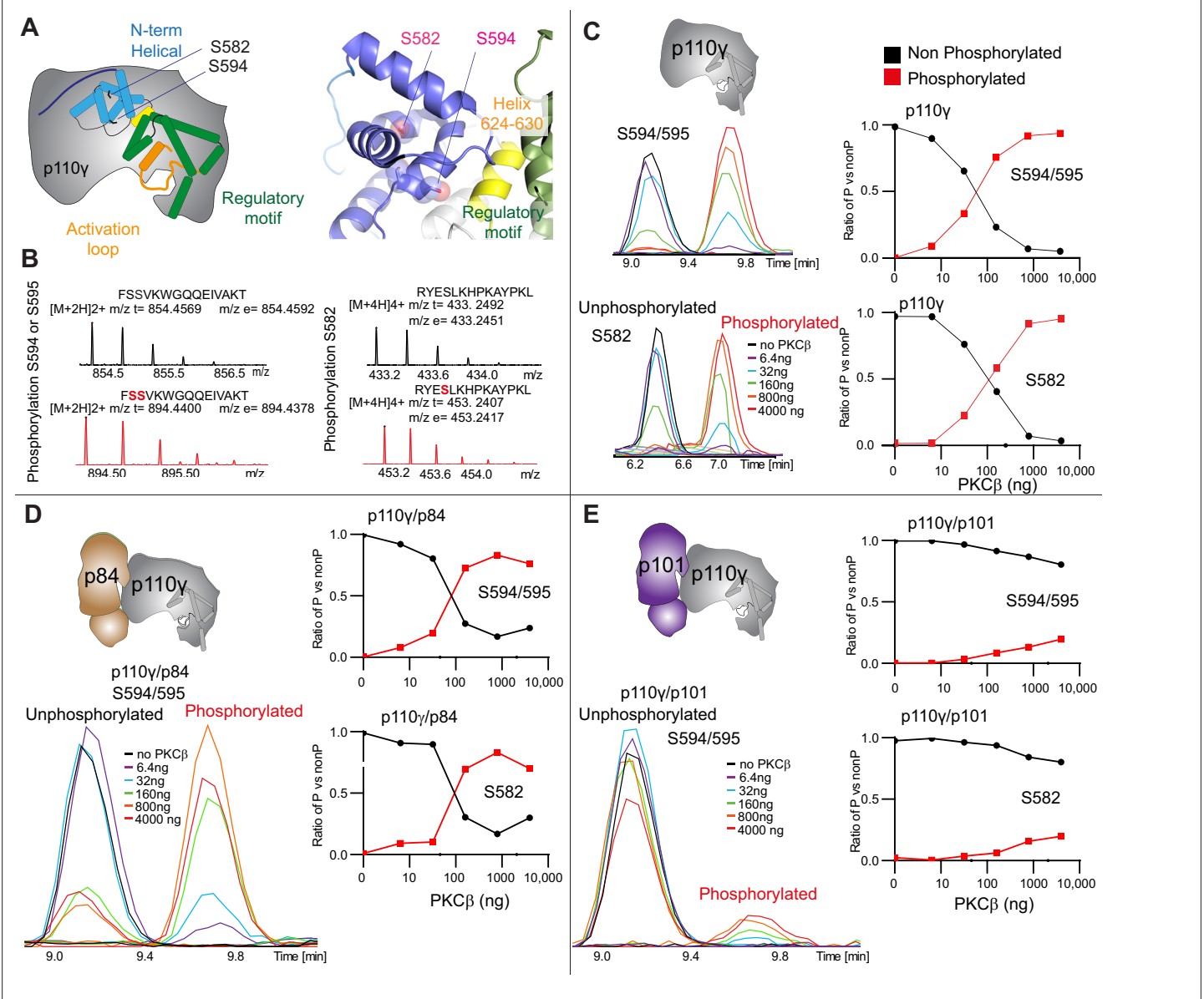

**Figure 3.** PKCβ leads to dual phosphorylation of internal sites in the helical domain, with selectivity for apo p110γ and p110γ-p84 over p110γ-p101. (**A**) Putative phosphorylation sites mapped on the structure of p110γ (PDB: 7MEZ) and cartoon schematic. The regions are colored based on domain schematics featured in *Figure 2A*. (**B**) Raw MS spectra of the unphosphorylated and phosphorylated peptide for a region spanning 579–592 (RYESLKHPKAYPKL) and 593–607 (FSSVKWGQQEIVAKT). The putative phosphorylation sites in the sequence are shown in red, with the m/z theoretical (m/z t) and m/z experimental (m/z t) shown below each sequence. (**C–E**) Extracted traces and ratios of the intensity of extracted ion traces of different phosphorylation site peptides (top to bottom: S594/S595 and S582) from (**C**) p110γ, (**D**) p110γ/p84, or (**E**) p110γ/p101 samples treated with increasing concentration of PKCβ according to the legend. The black traces in the ratio graphs are the intensity of the non-phosphorylated peptide, and the red traces in the ratio graphs are the intensity of the phosphorylated peptide.

The online version of this article includes the following figure supplement(s) for figure 3:

**Figure supplement 1.** MS/MS spectra of peptides spanning S582 and S594/S595 for both phosphorylated and unphosphorylated states.

This is indicative of cooperative unfolding of extended protein regions, with H/D exchange occurring faster than the refolding event. This region is where the PKCβ phosphorylation sites are located and may explain how the buried residues S582 and S594/S595 can be exposed to PKCβ. This is compatible with the observation that p110γ-p101 is protected from phosphorylation, as it does not show EX1 kinetics in this region, whereas both p110γ and p110γ-p84 do (*Rathinaswamy et al., 2021a*).

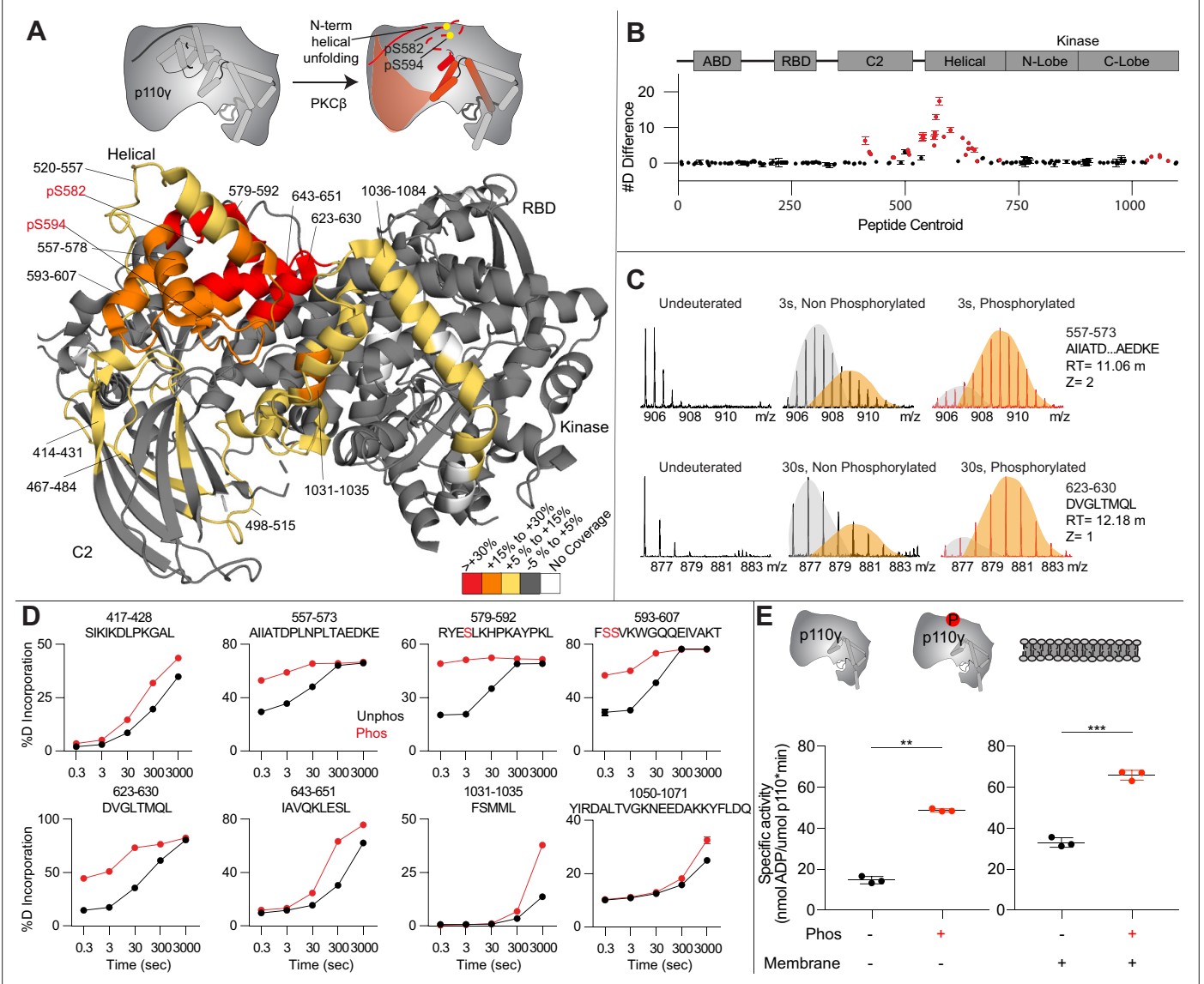

**Figure 4.** Activating phosphorylation at the helical domain leads to opening of the regulatory motif. (**A**) Hydrogen-deuterium exchange mass spectrometry (HDX-MS) comparing apo and phosphorylated p110γ. Significant differences in deuterium exchange are mapped on to the structure and cartoon of p110γ according to the legend (PDB: 7MEZ). (**B**) The graph of the #D difference in deuterium incorporation for p110γ, with each point representing a single peptide. Peptides colored in red are those that had a significant change in the mutants (greater than 0.4 Da and 5% difference at any time point, with a two tailed t-test p<0.01). Error bars are SD (n=3). (**C**) Representative bimodal distribution (EX1 kinetics) observed in the helical domain peptides of p110γ. (**D**) Representative p110γ peptides displaying increases in exchange in the phosphorylated state are shown. For all panels, error bars show SD (n = 3). (**E**) Measurement of ATP to ADP conversion of phosphorylated and non-phosphorylated p110γ (1000 nM final concentration) ATPase activity in the absence (left) and presence of PIP₂ membranes (5% phosphatidylinositol 4,5-bisphosphate [PIP₂], 95% phosphatidylserine [PS]) activation (right). Significance is indicated by **(<0.001%) and ***(<0.0001%).

When we compared phosphorylated p110γ (>90.8% as measured by mass spectrometry at both sites) to unphosphorylated p110γ we observed extensive increases in dynamics in the C2, helical domain, and kinase domain (*Figure 4A–D*). The largest increases in exchange upon phosphorylation were located in the N-terminal region of the helical domain, with the peptides directly adjacent to the phosphorylation site showing almost complete deuterium incorporation at the earliest time points of exchange. This is indicative of significant disruption of the alpha helical secondary structure in this region. When we examined the exchange profiles in this region, they still underwent EX1 kinetics (*Figure 4C*), however, phosphorylated p110γ was enriched in the more fully deuterated species. In addition to the regions in the helical domain, a portion of the regulatory motif of the kinase domain

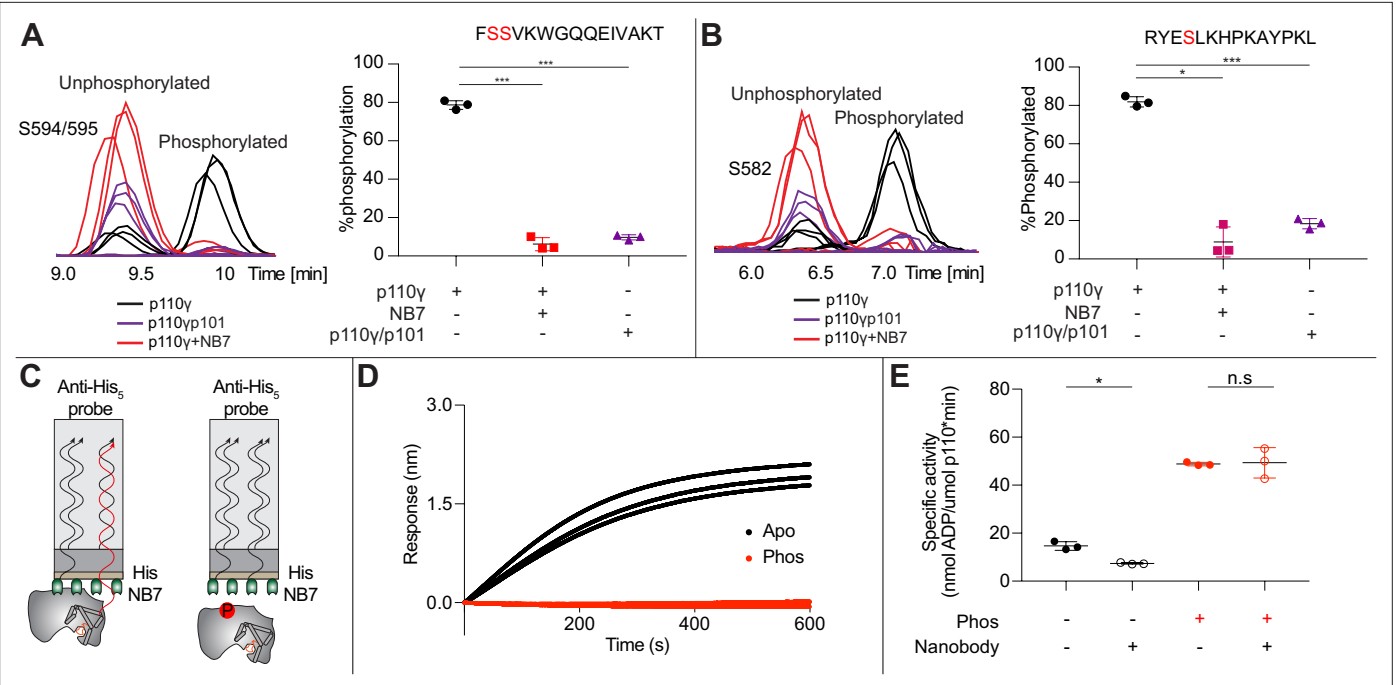

**Figure 5.** Nanobody NB7 blocks PKCβ phosphorylation, and phosphorylation prevents nanobody binding. (**A**) Extracted ion chromatograms for p110γ, p110γ-p101, and p110γ bound to NB7 are shown for the S594 or S595 phosphorylation site in p110γ. A bar graph showing the intensities of phosphorylated and non-phosphorylated p110γ peptide (593-607) for p110γ (black), p110γ with NB7 (red), and p110γ-p101 (purple) are shown to the right of the extracted ion chromatograms (n=3, right). In all experiments in panels A+B, PKCβ was present at 500 nM. Significance is indicated by ***(<0.0001%). (**B**) Extracted ion chromatograms for p110γ, p110γ-p101, and p110γ bound to NB7 are shown for the S582 phosphorylation site in p110γ. A bar graph showing the intensities of phosphorylated and non-phosphorylated p110γ peptide (579-592) p110γ (black), p110γ with NB7 (red), and p110γ-p101 (purple) are shown to the right of the extracted ion chromatograms (n=3, right). Significance is indicated by *(<0.01%) and ***(<0.0001%). The putative phosphorylation site is shown in red in the sequence above the bar graphs for both panels A+B. (**C**) Cartoon schematic of biolayer interferometry (BLI) analysis of the binding of immobilized His-NB7 to phosphorylated and non-phosphorylated p110γ. (**D**) Association curves for phosphorylated and non-phosphorylated p110γ (25 nM) binding to His-NB7 are shown (n=3). (**E**) ATPase kinase activity assays comparing the activation/inhibition of phosphorylated and non-phosphorylated p110γ (1000 nM) with or without nanobody (3000 nM final) in the absence of PIP₂ membranes. Significance is indicated by *(<0.05%) and NS (>0.05%).

also showed increased deuterium exposure. This included the k9-k12 helices that surround the activation loop of p110γ. These increases in exchange were similar to those we had observed in a R1021C oncogenic activating mutant of *PIK3CG* (*Rathinaswamy et al., 2021c*).

To further explore the potential role of phosphorylation in mediating p110γ activity, we examined the kinase activity of p110γ under two conditions: basal ATP turnover, and with PIP₂ containing lipid membranes. The experiments in the absence of PIP₂ measure turnover of ATP into ADP and phosphate and is a readout of basal catalytic competency. Experiments with PIP₂ measured ATP consumed in the generation of PIP₃, as well as in non-productive ATP turnover. The p110γ enzyme in the absence of stimulators is very weakly active toward PIP₂ substrate with only ~2-fold increased ATP turnover compared to in the absence of membranes. This is consistent with very weak membrane recruitment of p110γ complexes in the absence of lipid activators (*Rathinaswamy et al., 2023*). PKCβ-mediated phosphorylation enhanced the ATPase activity of p110γ ~2-fold in both the absence and presence of membranes (*Figure 4E*). This suggests that the effect of phosphorylation is to change the intrinsic catalytic efficiency of phosphorylated p110γ, with limited effect on membrane binding.

## Nanobody decreases p110γ phosphorylation

As NB7 bound at the interface of the helical and kinase domains that is exposed upon PKCβ phosphorylation of p110γ, we hypothesized that the nanobody would likely alter phosphorylation. We carried out PKCβ phosphorylation of p110γ, p110γ-p101, and p110γ bound to NB7. The presence of NB7 showed a significant decrease in p110γ phosphorylation at both sites (*Figure 5A–B*). We also wanted to determine whether p110γ phosphorylation reciprocally perturbed NB7 binding. BLI experiments

showed that there was no detectable binding of NB7 to phosphorylated p110γ (*Figure 5C–D*), consistent with phosphorylation disrupting the N-terminal region of the p110γ helical domain. In addition, lipid kinase assays using phosphorylated p110γ showed no detectable difference in activity when measured in the absence and presence of nanobody (*Figure 5E*), consistent with NB7 being unable to bind to phosphorylated p110γ.

## Discussion

Here, we find that the helical domain is a central regulator of the p110γ catalytic subunit of class IB PI3Kγ, with modulation of helical dynamics through binding partners or PTMs able to either increase or decrease lipid kinase activity. These results expand on previous work defining the helical domain as a central regulator of class IA PI3Ks, where the nSH2 domain of the p85 regulatory subunits makes inhibitory interactions that significantly inhibit lipid kinase activity of all class IA catalytic subunits (p110α, p110β, and p110δ) (*Mandelker et al., 2009*; *Miled et al., 2007*; *Burke and Williams, 2013*; *Burke, 2018*). This inhibitory interaction in class IA PI3Ks is disrupted in human cancers (helical hotspot

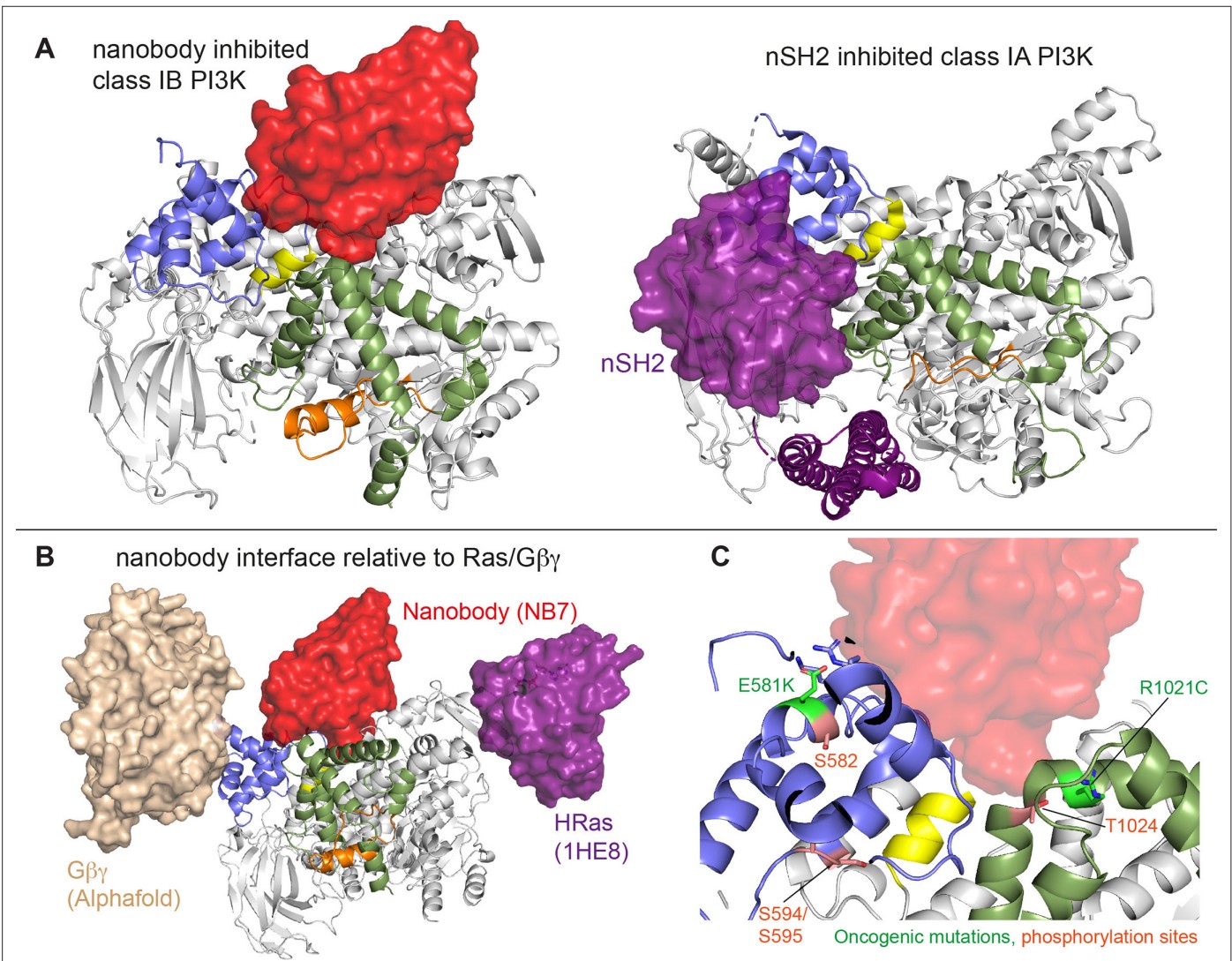

**Figure 6.** Comparison of nanobody binding site compared to p85 inhibition of class IA phosphoinositide 3 kinases (PI3Ks) and class IB activation sites. (**A**) Comparison of the nanobody NB7 binding site in p110γ compared to the nSH2 inhibitory site in p110α (PDB: 3HHM) (*Mandelker et al., 2009*). (**B**) Comparison of the nanobody NB7 binding site in p110γ compared to the X-ray structure of the Ras binding site (PDB: 1HE8) (*Pacold et al., 2000*) and the Alphafold model of Gβγ bound to p110γ (*Rathinaswamy et al., 2023*). (**C**) Oncogenic mutations and post-translational modifications in spatial proximity to the nanobody binding site.

mutations in *PIK3CA*) (*Samuels et al., 2004*) and immune disorders (helical mutations in *PIK3CD* in APDS1) (*Angulo et al., 2013*; *Lucas et al., 2014*). Class IB PI3Ks are uniquely compared to class IA PI3Ks, as they are not inhibited by p101 and p84 regulatory subunits, but instead potentiate GPCR activation. This lack of inhibition is due to the distinct binding interface of class IB PI3K regulatory subunits compared to class IA regulatory subunits, with only class IA regulatory subunits making direct inhibitory interactions with the kinase and helical domains of p110 catalytic subunits (*Rathinaswamy et al., 2021a*). Here, we show that a unique surface at the interface of the helical and kinase domains of p110γ is a potential site for the development of novel allosteric inhibitors that modulate p110γ activity.

The previously identified inhibitory nanobody (NB7) (*Rathinaswamy et al., 2021b*) bound with high affinity and inhibited all complexes of p110γ. The nanobody interface is distinct from how the nSH2 inhibits class IA PI3K activity, as its binding site is on the opposite face of the helical domain (*Figure 6A*). The mechanism of inhibition is also distinct, as the nSH2-helical interaction plays a critical role in preventing membrane recruitment of inhibited class I PI3Ks, with removal of this interface either through pYXXM motif binding or oncogenic mutations leading to increased membrane recruitment (*Burke et al., 2012*; *Burke et al., 2011*; *Zhang et al., 2011*). Analysis of the nanobody binding site compared to the structure of HRas-p110γ or the HDX-MS supported Alphafold-multimer prediction of Gβγ-p110γ (*Rathinaswamy et al., 2023*) shows that nanobody binding does not sterically block complex formation (*Figure 6B*). This is consistent with it not blocking membrane recruitment by Ras/Gβγ. The nanobody inhibited ATP turnover both in solution and on membranes, suggesting that it prevents formation of a catalytically competent conformation of p110γ, but still allows for membrane recruitment. Further development of small molecule allosteric binding partners in this allosteric pocket between the kinase and helical domain may reveal the specific molecular interactions in this pocket that mediate inhibition.

Oncogenic mutations are frequent in the class IA PI3K encoded by *PIK3CA*, with this being the second most frequently mutated gene in human cancer (*Lawrence et al., 2014*). Mutations in p110γ encoded by *PIK3CG* in cancer are less frequent, however, they can still provide insight into regulatory mechanisms that control activity. Oncogenic mutations in the kinase domain (R1021C) and helical domain (E581K) are in close proximity to the nanobody binding site, and both would be expected to disrupt the stability of the helical domain or regulatory motif of the kinase domain (*Figure 6C*). In addition to these mutations there are also multiple post-translational modifications that occur in this region, including inhibitory phosphorylation at T1024 (*Perino et al., 2011*) and activating phosphorylation at S582 (*Walser et al., 2013*). PKCβ is activated downstream of the IgE receptor in mast cells (*Walser et al., 2013*), but the full details of how this activates PI3Kγ have been unclear. We identified an additional PKCβ phosphorylation site located in the helical domain (S594/S595) (*Figure 6C*). Both the S582 and S594/S595 sites are not surface accessible and would require a transient opening of the helical domain for kinase accessibility. HDX-MS analysis of the helical domain of p110γ has shown that it is more dynamic than other class I PI3K isoforms (*Burke and Williams, 2013*; *Walser et al., 2013*), with the presence of the p101 regulatory subunit dramatically decreasing helical domain dynamics (*Vadas et al., 2013*). This putative mechanism of helical domain dynamics driving PKCβ phosphorylation is consistent with our observation that p101 subunits decreased p110γ phosphorylation >100-fold. PKCβ phosphorylation of p110γ leads to increased dynamics in both the helical and kinase domains with increased kinase activity, although only weakly compared to full activation by either membrane localized Ras or Gβγ. This increase was observed with both membrane and soluble substrate, so likely is not driven by altered membrane recruitment.

Overall, our biophysical and biochemical analysis of modulators of helical domain dynamics reveal the critical role of this domain in regulating class IB PI3Kγ activity. This raises possibilities for development of small molecule modulators that may either increase or decrease helical domain dynamics, leading to either activation or inhibition. The high-resolution structure of an allosteric inhibitor nanobody provides initial insight into which pockets can specifically be targeted. Multiple ATP competitive p110γ selective inhibitors are in clinical trials for human cancers (*Li et al., 2021*), with many having significant side effects. The identification of novel inhibitory strategies provides new opportunities for targeting p110γ dysregulation in human disease.

## Methods

### Plasmid generation

Plasmids encoding *Homo sapiens* p110γ (human), *Mus musculus* p84 (mouse), *Sus scrofa* p101 (porcine), and Gβγ were used as previously described (*Rathinaswamy et al., 2023*). The plasmids encoding the class IA PI3Ks were also used as previously described (*Dornan et al., 2017*; *Siempelkamp et al., 2017*). The pDONR223-PRKCB2 (PKCβII, uniprot identifier: P05771-2) was a gift from William Hahn & David Root (Addgene plasmid #23746; http://n2t.net/addgene: 23746; RRID:Addgene_23746) (*Johannessen et al., 2010*). The PKCβII construct contains an internal TEV site that cleaves the catalytic domains from the C1/C2 regulatory domains (TEV site inserted between residues 320 and 321 of PKCβ). This construct was subcloned into a pACEBAC Sf9 expression vector for Sf9 protein production. All constructs were cloned to include a 10× histidine tag, a 2× strep tag, and a tobacco etch virus protease cleavage site on the N-terminus. For p110γ and PKCβII this tag was included at the N-terminus, with this tag included at the N-terminus of either p84 or p101 for purification of p110γ-p101 and p110γ-p84. Full details of the plasmids are included in Appendix 1—key resources table.

### Virus generation and amplification

The plasmids encoding genes for insect cell expression were transformed into DH10MultiBac cells (MultiBac, Geneva Biotech) to generate bacmid containing the genes of interest. Successful generation was identified by blue-white colony screening and the bacmid was purified using a standard isopropanol-ethanol extraction method. Bacteria were grown overnight (16 hr) in 3–5 mL 2xYT (BioBasic #SD7019). Cells were spun down and the pellet was resuspended in 300 μL of 50 mM Tris-HCl, pH 8.0, 10 mM EDTA, 100 mg/mL RNase A. The pellet was lysed by the addition of 300 μL of 1% sodium dodecyl sulfate (SDS) (wt/vol), 200 mM NaOH, and the reaction was neutralized by addition of 400 μL of 3.0 M potassium acetate, pH 5.5. Following centrifugation at 21130 RCF and 4°C (Rotor #5424R), the supernatant was mixed with 800 μL isopropanol to precipitate bacmid DNA. Following centrifugation, the pelleted bacmid DNA was washed with 500 μL 70% ethanol three times. The pellet was then air dried for 1 min and re-suspended in 50 μL Buffer EB (10 mM Tris-Cl, pH 8.5; all buffers from QIAprep Spin Miniprep Kit, QIAGEN #27104). Purified bacmid was then transfected into Sf9 cells. Two mL of Sf9 cells at $0.6 \times 10^6$ cells/mL were aliquoted into a six-well plate and allowed to attach to form a confluent layer. Transfection reactions were prepared mixing 8–12 μg of bacmid DNA in 100 μL 1×PBS and 12 μg polyethyleneimine (Polyethyleneimine 'Max' MW 40.000, Polysciences #24765, USA) in 100 μL 1×PBS and the reaction was allowed to proceed for 20–30 min before addition to an Sf9 monolayer containing well. Transfections were allowed to proceed for 5–6 days before harvesting virus containing supernatant as a P1 viral stock.

Viral stocks were further amplified by adding P1 to Sf9 cells at $\sim 2 \times 10^6$ cells/mL (2/100 volume ratio). This amplification was allowed to proceed for 4–5 days and resulted in a P2 stage viral stock that was used in final protein expression. Harvesting of P2 viral stocks was carried out by centrifuging cell suspensions in 50 mL Falcon tubes at 2281 RCF (Beckman GS-15). To the supernatant containing virus, 5–10% inactivated fetal bovine serum (FBS; VWR Canada #97068-085) was added and the stock was stored at 4°C.

### Expression and purification of PI3Kγ, PI3Kα/β/δ, and PKCβ constructs

PI3Kγ *and PKCβ* constructs were expressed in Sf9 insect cells using the baculovirus expression system. Following 55 hr after infection with P2 viral stocks, cells were harvested by centrifuging at 1680 RCF (Eppendorf Centrifuge 5810R) and the pellets were snap-frozen in liquid nitrogen. The complex was purified through a combination of nickel affinity, streptavidin affinity, and size-exclusion chromatography.

Frozen insect cell pellets were resuspended in lysis buffer (20 mM Tris pH 8.0, 100 mM NaCl, 10 mM imidazole pH 8.0, 5% glycerol [vol/vol], 2 mM ME), protease inhibitor (Protease Inhibitor Cocktail Set III, Sigma) and sonicated for 2 min (15 s on, 15 s off, level 4.0, Misonix sonicator 3000). Triton X was added to the lysate to a final concentration of 0.1% and clarified by spinning at 15,000 RCF at 4°C for 45 min (Beckman Coulter JA-20 rotor). The supernatant was loaded onto a 5 mL HisTrap FF crude column (GE Healthcare) equilibrated in NiNTA A buffer (20 mM Tris pH 8.0, 100 mM NaCl, 20 mM imidazole pH 8.0, 5% [vol/vol] glycerol, 2 mM ME). The column was washed with high salt NiNTA A

buffer (20 mM Tris pH 8.0, 1 M NaCl, 20 mM imidazole pH 8.0, 5% [vol/vol] glycerol, 2 mM ME), NiNTA A buffer, 6% NiNTA B buffer (20 mM Tris pH 8.0, 100 mM NaCl, 250 mM imidazole pH 8.0, 5% [vol/vol] glycerol, 2 mM ME) and the protein was eluted with 100% NiNTA B. The eluent was loaded onto a 5 mL StrepTrap HP column (GE Healthcare) equilibrated in gel filtration buffer (20 mM Tris pH 8.5, 100 mM NaCl, 50 mM ammonium sulfate and 0.5 mM tris(2-carboxyethyl) phosphine [TCEP]). To purify *PI3Kα/β/δ*, the purification protocol was performed as described above but instead the protein was eluted in PI3Kα gel filtration buffer (20 mM HEPES 7.5, 150 mM NaCl, 0.5 mM TCEP). The column was washed with the corresponding gel filtration buffer and loaded with tobacco etch virus protease. After cleavage on the column overnight, the PI3K protein constructs were eluted in gel filtration buffer. The protein was concentrated in a 50,000 MWCO Amicon Concentrator (Millipore) to <1 mL and injected onto a Superdex 200 10/300 GL Increase size-exclusion column (GE Healthcare) equilibrated in gel filtration buffer. After size exclusion, the protein was concentrated, aliquoted, frozen, and stored at –80°C. For PKCβ, the protein was eluted from the strep column in gel filtration buffer, and the eluate was then loaded on a 1 mL HisTrap FF column to remove his tagged LipTev. The flowthrough was collected, and the column was washed with 2 mL of gel filtration buffer. These fractions were pooled and concentrated and stored at –80°C.

To purify phosphorylated p110γ, the purification protocol as described above was performed but PKCβ was added to the strep column at a molar ratio of 1:3 (PKCβ:p110γ) along with LipTEV, 20 mM MgCl$_2$, and 1 mM ATP and allowed to incubate on ice for 4 hr. The protein was eluted by adding 7 mL of gel filtration buffer and treated with a second dose of PKCβ (same ratio as above) and allowed to incubate on ice for another 3 hr. For non-phosphorylated p110γ, same protocol was followed with the exception in the addition of PKCβ. Both the proteins were concentrated in a 50,000 MWCO Amicon Concentrator (Millipore) to <1 mL and injected onto a Superdex 200 10/300 GL Increase size-exclusion column (GE Healthcare) equilibrated in gel filtration buffer. The final phosphorylation level of the two sites was characterized by mass spectrometry, with these values being 92% and 90.8%, for S582 and S594/S595, respectively. After size exclusion, the protein was concentrated, aliquoted, frozen, and stored at –80°C.

## Expression and purification of lipidated G for kinase activity assays

Full-length, lipidated human *Gβγ* (Gβ1γ2) was expressed in Sf9 insect cells and purified as described previously. After 65 hr of expression, cells were harvested, and the pellets were frozen as described above. Pellets were resuspended in lysis buffer (20 mM HEPES pH 7.7, 100 mM NaCl, 10 mM βME, protease inhibitor [Protease Inhibitor Cocktail Set III, Sigma]) and sonicated for 2 min (15 s on, 15 s off, level 4.0, Misonix sonicator 3000). The lysate was spun at 500 RCF (Eppendorf Centrifuge 5810R) to remove intact cells and the supernatant was centrifuged again at 25,000 RCF for 1 hr (Beckman Coulter JA-20 rotor). The pellet was resuspended in lysis buffer and sodium cholate was added to a final concentration of 1% and stirred at 4°C for 1 hr. The membrane extract was clarified by spinning at 10,000 RCF for 30 min (Beckman Coulter JA-20 rotor). The supernatant was diluted three times with NiNTA A buffer (20 mM HEPES pH 7.7, 100 mM NaCl, 10 mM imidazole, 0.1% C12E10, 10 mM ME) and loaded onto a 5 mL HisTrap FF crude column (GE Healthcare) equilibrated in the same buffer. The column was washed with NiNTA A, 6% NiNTA B buffer (20 mM HEPES pH 7.7, 25 mM NaCl, 250 mM imidazole pH 8.0, 0.1% C12E10, 10 mM ME) and the protein was eluted with 100% NiNTA B. The eluent was loaded onto HiTrap Q HP anion exchange column equilibrated in Hep A buffer (20 mM Tris pH 8.0, 8 mM CHAPS, 2 mM dithiothreitol [DTT]). A gradient was started with Hep B buffer (20 mM Tris pH 8.0, 500 mM NaCl, 8 mM CHAPS, 2 mM DTT) and the protein was eluted in ~50% Hep B buffer. The eluent was concentrated in a 30,000 MWCO Amicon Concentrator (Millipore) to <1 mL and injected onto a Superdex 75 10/300 GL size-exclusion column (GE Healthcare) equilibrated in Gel Filtration buffer (20 mM HEPES pH 7.7, 100 mM NaCl, 10 mM CHAPS, 2 mM TCEP). Fractions containing protein were pooled, concentrated, aliquoted, frozen, and stored at –80°C.

## Expression and purification of nanobody

Nanobody NB7-PIK3CG with a C-terminal 6× His tag was expressed from a pMESy4 vector in the periplasm of WK6 *E. coli*. A 1 L culture was grown to OD600 of 0.7 in Terrific Broth containing 0.1% glucose and 2 mM MgCl$_2$ in the presence of 100 μg/mL ampicillin and was induced with 0.5 mM isopropyl-β-D-thiogalactoside (IPTG). Cells were harvested the following day by centrifuging at 2500

RCF (Eppendorf Centrifuge 5810R) and the pellet was snap-frozen in liquid nitrogen. The frozen pellet was resuspended in 15 mL of TES buffer containing 200 mM Tris pH 8.0, 0.5 mM ethylenediaminetetraacetic acid (EDTA), and 500 mM sucrose and was mixed for 45 min at 4°C. To this mixture, 30 mL of TES buffer diluted four times in water was added and mixed for 45 min at 4°C to induce osmotic shock. The lysate was clarified by centrifuging at 14,000 rpm for 15 min (Beckman Coulter JA-20 rotor). Imidazole was added to the supernatant to final concentration of 10 mM loaded onto a 5 mL HisTrap FF crude column (GE Healthcare) equilibrated in NiNTA A buffer (20 mM Tris pH 8.0, 100 mM NaCl, 20 mM imidazole pH 8.0, 5% [vol/vol] glycerol, 2 mM β-mercaptoethanol [ME]). The column was washed with high salt NiNTA A buffer (20 mM Tris pH 8.0, 1 M NaCl, 20 mM imidazole pH 8.0, 5% [vol/vol] glycerol, 2 mM ME), followed by 100% NiNTA A buffer, then a 6% NiNTA B wash buffer (20 mM Tris pH 8.0, 100 mM NaCl, 250 mM imidazole pH 8.0, 5% [vol/vol] glycerol, 2 mM ME) and the protein was eluted with 100% NiNTA B. The eluent was concentrated in a 10,000 MWCO Amicon Concentrator (Millipore) to <1 mL and injected onto a Superdex 75 10/300 GL Increase size-exclusion column (GE Healthcare) equilibrated in gel filtration buffer (20 mM Tris pH 8.5, 100 mM NaCl, 50 mM ammonium sulfate, and 0.5 mM TCEP). Following size exclusion, the protein was concentrated, frozen, and stored at –80°C.

## Lipid vesicle preparation for kinase activity assays

Lipid vesicles containing 5% brain phosphatidylinositol 4,5-bisphosphate (PIP2) and 95% brain phosphatidylserine (PS) were prepared by mixing the lipids solutions in organic solvent. The solvent was evaporated in a stream of argon following which the lipid film was desiccated in a vacuum for 45 min. The lipids were resuspended in lipid buffer (20 mM HEPES pH 7.0, 100 mM NaCl, and 10% glycerol) and the solution was vortexed for 5 min followed by sonication for 15 min. The vesicles were then subjected to 10 freeze thaw cycles and extruded 11 times through a 100 nm filter (T&T Scientific: TT-002-0010). The extruded vesicles were sub-aliquoted and stored at –80°C. Final vesicle concentration was 2 mg/mL.

## Kinase assays

All kinase assays were done using Transcreener ADP2 Fluorescence Intensity (FI) assays (Bellbrook labs) which measures ADP production. All assays contained ATP at a final concentration of 100 µM, and those with membranes used vesicles containing 5% $PIP_2$ and 95% PS at a final concentration of 0.5 mg/mL.

For assays measuring the inhibition by nanobody, 4× kinase (final concentration: 330 nM for p110γ, 300 nM for p110γ/p84, and 12 nM for p110γ/p101) was mixed with varying 4× concentrations of nanobody (final concentration: 2 µM to 2.7 nM) or kinase buffer (20 mM HEPES pH 7.5, 100 mM NaCl, 3 mM MgCl₂, 0.03% CHAPS, 2 mM TCEP, and 1 mM EGTA) and allowed to sit on ice for 15 min. Two µL of protein mix was mixed with 2 µL of lipid solution containing Gβγ (1 µM final concentration), ATP (100 µM), $PIP_2$ lipid vesicles (0.5 mg/ml final concentration), and lipid buffer (25 mM HEPES pH 7, 5% glycerol, and 100 mM NaCl) and incubated at 20°C for 60 min.

For assays comparing the difference in activation between phosphorylated and non-phosphorylated p110γ, 2× kinase (final concentrations: 1 µM) was mixed with 2× lipid solutions containing ATP (100 µM), and lipid buffer and either nanobody (3 µM final concentration), $PIP_2$ lipid vesicles (0.5 mg/mL final concentration), or both nanobody and lipid. The reaction was incubated at 20°C for 60 min.

After the 60 min incubation, all reactions were stopped with 4 µL of 2× stop and detect solution containing Stop and Detect buffer (20 mM HEPES, 0.02% Brij-35, 40 mM EDTA pH 7.5), 8 nM ADP Alexa Fluor 594 Tracer and 93.7 µg/mL ADP2 Antibody IRDye QC-1, covered and incubated at 20°C for 1 hr before reading the fluorescence. The fluorescence intensity was measured using a SpectraMax M5 plate reader at excitation 590 nm and emission 620 nm. All data was normalized against the appropriate measurements obtained for 100 µM ATP and 100 µM ADP with no kinase. The percent ATP turnover was interpolated using a standard curve (0.1–100 µM ADP). Interpolated values were then used to calculate the specific activity of the enzyme.

## Biolayer interferometry

All BLI experiments were performed using the Octet K2 instrument (Fortebio Inc). For all experiments His-tagged nanobody (500 nM) was immobilized on an Anti-Penta-His biosensor for 600 s, and

the sensor was dipped into varying concentrations of the protein complex being measured. A dose response was carried out for p110γ, p110γ-p84, and p110γ-p101 (50 nM – 1.9 nM), with association occurring for 600 s, followed by a 1200s dissociation in Octet Buffer (20 mM Tris pH 8.5, 100 mM NaCl, 50 mM ammonium sulfate, 0.1% bovine serum albumin, and 0.02% Tween 20). Experiments comparing class IA PI3K versus class IB PI3Kγ used 50 nM of each class IA PI3K.

When comparing nanobody binding to phosphorylated and unphosphorylated p110γ, we used a final concentration of 25 nM for both phosphorylated and non-phosphorylated p110γ with association occurring for 600 s, followed by a 600 s dissociation. The $K_D$ (dissociation constant) for the different p110γ complexes was calculated from the binding curves based on their global fit to a 1:1 binding model using ForteBio data analysis 12.0 (Fortebio Inc).

## SLB TIRF microscopy experiments

The membrane binding dynamics of Dy647-p84-p110γ were measured in the absence and presence of nanobody 7 (NB7) using TIRF microscopy. As previously described (*Rathinaswamy et al., 2023*), SLBs were formed using 50 nm extruded small unilamellar vesicles containing the following lipids: 1,2-dioleoyl-*sn*-glycero-3-phosphocholine (18:1 DOPC, Avanti #850375C), 1,2-dioleoyl-*sn*-glycero-3-phospho-L-serine (18:1 DOPS, Avanti #840035C), 1,2-dioleoyl-*sn*-glycero-3-phosphoethanolamine-*N*-[4-(p-maleimidomethyl)cyclohexane-carboxamide] (18:1 MCC-PE, Avanti #780201C). Lipid compositions reported in figure legends represent the molar percentage of each lipid species.

To create SLBs, a total concentration of 0.25 mM lipids was solvated in 1× PBS (pH 7.4) and deposited on Piranha etched glass coverslips (25×75 mm) adhered to an IBIDI chamber. After a 30 min incubation, membranes were washed with 4 mL of 1× PBS (pH 7.4) and then blocked for 10 min with 1 mg/mL beta casein (Thermo Fisher Science, Cat#37528) in 1× PBS (pH 7.4) (Corning, Cat#46-013CM). To conjugate H-Ras to maleimide lipids (MCC-PE), blocked membranes were incubated with 30 μM H-Ras (GDP) in buffer containing 1× PBS (pH 7.4), 1 mM MgCl$_2$, 50 μM GDP, and 0.1 mM TCEP for 2 hr. The membrane conjugation reaction was terminated after 2 hr with 1× PBS (pH 7.4) containing 5 mM β-ME. Membranes were then washed and stored in 1× PBS (pH 7.4) until performing the TIRF microscopy (TIRF-M) membrane binding experiments. H-Ras was purified as previously described (*Rathinaswamy et al., 2023*).

To perform the TIRF-M membrane binding assays, 200 nM farnesyl-*Gβγ* was equilibrated into the supported membranes for 30 min. In parallel, nucleotide exchange of H-Ras (GDP) was performed by adding 50 nM guanine nucleotide exchange factor (SosCat) in 1× PBS (pH 7.4), 1 mM MgCl$_2$, 50 μM GDP. To measure membrane binding, Dy647-p84-p110γ was diluted into the following buffer: 20 mM HEPES (pH 7.0), 150 mM NaCl, 50 μM GTP, 1 mM ATP, 5 mM MgCl$_2$, 0.5 mM EGTA, 20 mM glucose, 200 μg/mL beta casein (Thermo Scientific, Cat#37528), 20 mM BME, 320 μg/mL glucose oxidase (Serva, #22780.01 *Aspergillus niger*), 50 μg/mL catalase (Sigma, #C40-100MG Bovine Liver), and 2 mM Trolox. Trolox was prepared as previously described (*Hansen et al., 2019*). Perishable reagents (i.e., glucose oxidase, catalase, and Trolox) were added 10 min before image acquisition.

TIRF-M experiments were performed using an inverted Nikon Ti2 microscope with a 100× Nikon (1.49 NA) oil immersion objective. The x-axis and y-axis positions were controlled using a Nikon motorized stage. Dy647-p84-p110γ was excited with a 637 nm diode laser (OBIS laser diode, Coherent Inc Santa Clara, CA, USA) controlled with an acousto-optic tunable filter and laser launch built by Vortran (Sacramento, CA, USA). The power output measured through the objective for single particle imaging was 1–3 mW. Excitation light passing through quad multi-pass dichroic filter cube (Semrock). Fluorescence emission passed through Nikon emission filter wheel containing the following 25 mm ET700/75 M emission filters (Semrock) before being detected on iXion Life 897 EMCCD camera (Andor Technology Ltd., UK). All TIRF-M experiments were performed at room temperature (23°C). Microscope hardware was controlled using Nikon NIS elements. Data analysis was performed using ImageJ/Fiji and Prism graphing program.

## Cryo-EM sample preparation and data collection

Three μL of purified nanobody-bound p110γ at 0.45 mg/mL was adsorbed onto C-Flat 2/2-T grids that were glow discharged for 25 s at 15 mA. Grids were then plunged into liquid ethane using a Vitrobot Mark IV (Thermo Fisher Scientific) with the following settings: –5 blot force, 1.5 s blot time, 100% humidity, and 4°C. Vitrified specimens were screened for ice and particle quality at the UBC

high-resolution macromolecular electron microscopy (HRMEM) facility using a 200 kV Glacios transmission electron microscope equipped with a Falcon 3EC direct electron detector (DED). Clipped grids were sent to the Pacific Northwest Cryo-EM Center (PNCC) where 7322 movies were collected using a Titan Krios equipped with a Gatan K3 DED and a BioQuantum K3 energy filter with a slit width of 20 eV. The movies were collected at a physical pixel size of 0.830 Å/pix and a total dose of 50 e⁻/$Å^2$ over 50 frames.

## Cryo-EM image analysis

The data were processed using cryoSPARC v.3.3.2 (*Punjani et al., 2017*). The movies were pre-processed by patch motion correction using default settings except Fourier cropping by a factor of 2, followed by patch CTF estimation using default settings. A 3D map of PI3K p110γ-p101 complex (EMD-23808) was used to create 2D projections for use as templates to auto-pick 1,463,553 particles. Particles were extracted with a box size of 380 pixels, Fourier cropped to a box size of 96 pixels and subjected to 2D classification. After discarding classes with obvious noise and no features, 795,162 particles were used for multiple rounds of ab initio reconstruction and heterogeneous refinement using four or five classes; 365,178 particles, which generated the two best 3D reconstruction, were used to carry out per-particle local-motion correction with 760 pixels box size later downsized to 380 pixels followed by several rounds of ab initio reconstruction and heterogeneous refinement using three or five classes; and 149,603 from best class were further refined by homogeneous refinement and a final non-uniform-refinement which generated a reconstruction with an overall resolution of 3.02 Å based on the Fourier shell correlation 0.143 criterion.

## Building the structural model of p110γ-NB7

The previous structural model of full-length p110γ from the complex of p110γ-p101 (PDB: 7MEZ) (*Rathinaswamy et al., 2021a*) was fit into the map using Chimera (*Pettersen et al., 2004*). A model of the nanobody was generated using Alphafold2 using the Colabfold v1.5.2 server (*Mirdita et al., 2022*). The CDR loops were removed from this initial model, and the remaining nanobody was fit into the map using Chimera. The final structure was built by iterative rounds of automated model building in Phenix, manual model building in COOT (*Emsley et al., 2010*), and refinement in Phenix. real_space_refine using realspace, rigid body, and adp refinement with tight secondary structure restraints (*Afonine et al., 2012*). This allowed for unambiguous building of the CDRs of the nanobody, and their interface with p110γ. The full refinement and validation statistics are shown in *Supplementary file 1*.

## Phosphorylation analysis

For the dose-response phosphorylation of p110γ, p110γ/p84, and p110γ/p101, each protein or complex (750 nM) was mixed with ATP (200 µM), GFB (20 mM Tris pH 8.5, 100 mM NaCl, 50 mM ammonium sulfate and 0.5 mM TCEP), $MgCl_2$ (20 mM), and various amounts of PKCβ (4 µg, 800 ng, 160 ng, 32 ng, 6.4 ng, and 0 ng). Reactions were incubated for 3 hr on ice and quenched with 50 µL of ice-cold acidic quench buffer (0.7 M guanidine-HCl, 1% formic acid), followed by immediate freezing using liquid nitrogen and storage at −80°C.

For the experiment studying the effect of nanobody on phosphorylation, p110γ or p110γ/p101 (500 nM) was mixed with ATP (1 mM), GFB (20 mM Tris pH 8.5, 100 mM NaCl, 50 mM ammonium sulfate, and 0.5 mM TCEP), $MgCl_2$(20 mM), with nanobody and PKCβ present at 1200 nM and 500 nM, respectively. Reactions were incubated for 1 hr at room temperature and quenched with 54 µL of ice-cold acidic quench buffer (0.7 M guanidine-HCl, 1% formic acid) followed by immediate freezing using liquid nitrogen and storage at −80°C.

Phosphorylation of all proteins was confirmed using mass spectrometry and PEAKS7 analysis. The LC-MS analysis of these samples was carried out using the same pipeline as used in the HDX-MS section. The phosphorylated and non-phosphorylated peptide ratios were determined by generating extracted ion chromatograms for each phosphorylated or non-phosphorylated peptide using their molecular formula and charge state in the Bruker Compass Data Analysis software. The area under each extracted curve was then extracted. The full MS quantification of each of the phosphorylated and non-phosphorylated peptide is provided in *Source data 1*.

## Hydrogen-deuterium exchange mass spectrometry

Exchange reactions to assess differences in p110γ upon phosphorylation were carried out at 20°C in 10 μL volumes with final concentrations of 1.6 μM for both apo and phosphorylated p110γ. A total of two conditions were assessed: p110γ apo and PKCβ phosphorylated p110γ. The H/D exchange reaction was initiated by the addition of 8 μL $D_2O$ buffer (94.3% $D_2O$, 100 mM NaCl, 20 mM HEPES pH 7.5) to the 2 μL protein for a final $D_2O$ concentration of 75.4%. Exchange was carried out over five time points (3 s on ice, and 3 s, 30 s, 300 s, and 3000 s at 20°C) and the reaction was quenched with addition of 60 μL of ice-cold acidic quench buffer (0.7 M guanidine-HCl, 1% formic acid). After quenching, samples were immediately frozen in liquid nitrogen and stored at –80°C. All reactions were carried out in triplicate.

## Protein digestion and MS/MS data collection

Protein samples for both HDX-MS and phosphorylation analysis were analyzed using the same LC-MS setup. Samples were rapidly thawed and injected onto an integrated fluidics system containing an HDx-3 PAL liquid handling robot and climate-controlled chromatography system (LEAP Technologies), a Dionex Ultimate 3000 UHPLC system, as well as an Impact HD QTOF Mass spectrometer (Bruker). The protein was run over two immobilized pepsin columns (Applied Biosystems; Poroszyme Immobilized Pepsin Cartridge, 2.1 mm × 30 mm; Thermo Fisher 2-3131-00; at 10°C and 2°C respectively) at 200 μL/min for 3 min. The resulting peptides were collected and desalted on a C18 trap column (Acquity UPLC BEH C18 1.7 mm column [2.1×5 mm]; Waters 186003975). The trap was subsequently eluted in line with an ACQUITY 1.7 μm particle, 100×1 mm² C18 UPLC column (Waters 186002352), using a gradient of 3–35% B (buffer A, 0.1% formic acid; buffer B, 100% acetonitrile) over 11 min immediately followed by a gradient of 35–80% B over 5 min. MS experiments acquired over a mass range from 150 to 2200 mass/charge ratio (m/z) using an electrospray ionization source operated at a temperature of 200°C and a spray voltage of 4.5 kV.

## Peptide identification

Peptides were identified using data-dependent acquisition following tandem MS/MS experiments (0.5 s precursor scan from 150 to 2000 m/z; twelve 0.25 s fragment scans from 150 to 2000 m/z). MS/MS datasets were analyzed using PEAKS7 (PEAKS), and a false discovery rate was set at 0.1% using a database of purified proteins and known contaminants. The same approach was used to identify phosphorylated and non-phosphorylated peptides for our in vitro phosphorylation experiments, with variable phosphorylation of STY residues was added to the search. The search parameters were set with a precursor tolerance of 20 parts per million, fragment mass error 0.02 Da, and charge states from 1 to 8, with a selection criterion of peptides that had a −10logp score of >24.03 for phosphorylated and >23.05 for non-phosphorylated. The MS/MS spectra of the PKCβ phosphorylated peptides are included in *Figure 3—figure supplement 1*.

## Mass analysis of peptide centroids and measurement of deuterium incorporation

HD-Examiner Software (Sierra Analytics) was used to automatically calculate the level of deuterium incorporation into each peptide. All peptides were manually inspected for correct charge state, correct retention time, and appropriate selection of isotopic distribution. Deuteration levels were calculated using the centroid of the experimental isotope clusters. HDX-MS results are presented with no correction for back exchange shown in *Source data 1*, with the only correction being applied correcting for the deuterium oxide percentage of the buffer used in the exchange (75.4%). Changes in any peptide at any time point greater than specified cut-offs (5% and 0.45 Da) and with an unpaired, two-tailed t-test value of p<0.01 was considered significant. A number of peptides in the helical domain showed isotope distributions consistent with EX1 H/D exchange. Attempts to define the relative percentages of each population using HDExaminer were extremely noisy, so representative EX1 profiles are shown in *Figure 4C*. The raw peptide deuterium incorporation graphs for a selection of peptides with significant differences are shown in *Figure 4D*, with the raw data for all analyzed peptides in *Source data 1*. To allow for visualization of differences across all peptides, we utilized number of deuteron difference (#D) plots (*Figure 4B*). These plots show the total difference in deuterium incorporation

over the entire H/D exchange time course, with each point indicating a single peptide. The data analysis statistics for all HDX-MS experiments are in Appendix 1—key resources table according to the guidelines of *Masson et al., 2019*. The mass spectrometry proteomics data have been deposited to the ProteomeXchange Consortium via the PRIDE partner repository (*Perez-Riverol et al., 2022*) with the dataset identifier PXD040765.

## Acknowledgements

JEB is supported by the Canadian Institute of Health Research (CIHR, 168998), and the Michael Smith Foundation for Health Research (MSFHR, scholar 17686). CKY is supported by CIHR (FDN-143228, PJT-168907) and the Natural Sciences and Engineering Research Council of Canada (RGPIN-2018-03951). SDH is supported by an NSF CAREER Award (MCB-2048060). This research project was supported in part by the UBC High Resolution Macromolecular Cryo-Electron Microscopy Facility (HRMEM). A portion of this research was supported by NIH grant U24GM129547 and performed at the PNCC at OHSU and accessed through EMSL (grid.436923.9), a DOE Office of Science User Facility sponsored by the Office of Biological and Environmental Research. We appreciate help from Theo Humphreys and Rose Marie Haynes with data collection at PNCC. Competing Interests: The authors declare that they have no competing interests.

## Additional information

### Competing interests

John E Burke: JEB reports personal fees from Scorpion Therapeutics, Reactive therapeutics and Olema Oncology; and research grants from Novartis. The other authors declare that no competing interests exist.

### Funding

| Funder | Grant reference number | Author |
|---|---|---|
| Canadian Institutes of Health Research | FDN-143228 | Calvin K Yip |
| Canadian Institutes of Health Research | PJT-168907 | Calvin K Yip |
| Natural Sciences and Engineering Research Council of Canada | RGPIN-2018-03951 | Calvin K Yip |
| National Science Foundation | MCB-2048060 | Scott D Hansen |
| Canadian Institutes of Health Research | 168998 | John E Burke |
| Michael Smith Health Research BC | 17686 | John E Burke |

The funders had no role in study design, data collection and interpretation, or the decision to submit the work for publication.

### Author contributions

Noah J Harris, Conceptualization, Validation, Investigation, Methodology, Writing – original draft, Writing – review and editing; Meredith L Jenkins, Conceptualization, Data curation, Formal analysis, Validation, Investigation, Methodology, Writing – original draft, Writing – review and editing; Sung-Eun Nam, Formal analysis; Manoj K Rathinaswamy, Matthew AH Parson, Formal analysis, Investigation, Methodology; Harish Ranga-Prasad, Udit Dalwadi, Brandon E Moeller, Eleanor Sheeky, Investigation, Methodology; Scott D Hansen, Formal analysis, Validation, Investigation, Methodology, Writing – original draft, Project administration, Writing – review and editing; Calvin K Yip, John E Burke, Conceptualization, Formal analysis, Supervision, Funding acquisition, Validation,

Investigation, Visualization, Methodology, Writing – original draft, Project administration, Writing – review and editing

## Author ORCIDs

Meredith L Jenkins http://orcid.org/0000-0002-0685-5048
Matthew AH Parson https://orcid.org/0000-0001-6270-559X
Eleanor Sheeky https://orcid.org/0000-0002-1501-550X
Scott D Hansen https://orcid.org/0000-0001-7005-6200
Calvin K Yip http://orcid.org/0000-0003-1779-9501
John E Burke http://orcid.org/0000-0001-7904-9859

Reviewer #1 (Public Review): https://doi.org/10.7554/eLife.88058.3.sa1
Reviewer #2 (Public Review): https://doi.org/10.7554/eLife.88058.3.sa2
Author Response: https://doi.org/10.7554/eLife.88058.3.sa3

---

## Additional files

### Supplementary files

• Supplementary file 1. Cryo electron microscopy (cryo-EM) data collection, refinement, and validation statistics.

• Supplementary file 2. HDX-MS data collection and validation statistics.

• MDAR checklist

• Source data 1. Raw source data for figures.

### Data availability

The model of the nanobody bound p110γ has been deposited in the protein data bank (PDB) with the identifier 8DP0, with the EM data available at the EMDB with the identifier EMD-27627. The mass spectrometry proteomics data have been deposited to the ProteomeXchange Consortium via the PRIDE partner repository (*Perez-Riverol et al., 2022*) with the dataset identifier PXD040765. All source data underlying all data shown in the figures and figure supplements is available in *Source data 1*. All genetic material generated for this study is available from the lead contact by request.

The following datasets were generated:

| Author(s) | Year | Dataset title | Dataset URL | Database and Identifier |
|---|---|---|---|---|
| Burke JE | 2023 | Allosteric activation or inhibition of PI3Kγ mediated through conformational changes in the p110γ helical domain | https://www.ebi.ac.uk/pride/archive/projects/PXD040765 | PRIDE, PXD040765 |
| Burke JE, Sung SE, Rathinaswamy MK, Yip CK | 2023 | Structure of p110 gamma bound to the Ras inhibitory nanobody NB7 | https://www.rcsb.org/structure/8DP0 | RCSB Protein Data Bank, 8DP0 |
| Burke JE, Sung SE, Rathinaswamy MK, Yip CK | 2023 | Structure of p110 gamma bound to the Ras inhibitory nanobody NB7 | https://www.ebi.ac.uk/emdb/EMD-27627 | Electron Microscopy Data Bank, EMD-27627 |

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

# Appendix 1

## Appendix 1—key resources table

| Reagent type (species) or resource | Designation | Source or reference | Identifiers | Additional information |
|---|---|---|---|---|
| Strain, strain background (*Escherichia coli*) | *E. coli* XL10-GOLD KanR Ultracompetent Cells | Agilent | 200317 | |
| Strain, strain background (*Escherichia coli*) | BL21 *E. coli* C41 (DE3) RIPL | PMID: 24876499 | C41 | |
| Strain, strain background (*Spodoptera frugiperda*) | Sf9 Insect Cells | Expression systems | 94-001S | |
| Strain, strain background (*Escherichia coli*) | *E. coli* DH10EMBacY Competent Cells | Geneva Biotech | DH10EMBacY | |
| Recombinant DNA reagent | pMultiBac-Gβ1/Gγ2 | PMID:34452907 | pOP737 | |
| Recombinant DNA reagent | pACEBac1-hsp110γ | PMID:34452907 | MR30 | |
| Recombinant DNA reagent | pMultiBac-hsp110γ-ssp101 | PMID:34452907 | MR22 | |
| Recombinant DNA reagent | pMultiBac-hsp110γ-mmp84 | PMID:34452907 | MR24 | |
| Recombinant DNA reagent | pFastBac HRas G12V | PMID:34452907 | BS9 | |
| Recombinant DNA reagent | biGBac hsp110γ/ybbr-hsp84 | PMID:36842083 | HP28 | |
| Recombinant DNA reagent | biGBac hsp110γ/ybbr-hsp101 | PMID:36842083 | HP29 | |
| Recombinant DNA reagent | his6-GST-PrescissionProtease-SNAP-RBD(K65E) | PMID:34452907 | pSH936 | |
| Recombinant DNA reagent | his6TEV-HRas(1-184aa) C118S, C181S | PMID:34452907 | pSH414 | |
| Recombinant DNA reagent | his6-G2, SNAP-G1 (DUAL FastBac) | PMID:34452907 | pSH651 | |
| Recombinant DNA reagent | pACEBAC-PKCβII (internal tev cleavage site) | This paper | pMR56 | |
| Recombinant DNA reagent | pFASTBac p110α | PMID: 28515318 | pOV1181 | |
| Recombinant DNA reagent | pFASTBac p110β | PMID: 28515318 | pOV1182 | |
| Recombinant DNA reagent | pFASTBac p110δ | PMID: 28515318 | pOV1183 | |
| Recombinant DNA reagent | pFASTBac p85β | This paper | EX21 | |
| Sequence-based reagent | Fwd primer for amplifying KD of PKCII | Sigma | MR51F | GTATTTTCAGGGCgccggtaccACGACCAACACTGTCTCCAAATTTG |
| Sequence-based reagent | Rvs primer for amplifying KD of PKCII | Sigma | MR51R | gactcgagcggccgcTTATAGCTCTTGACTTCGGGTTTTAAAAATTCAG |
| Sequence-based reagent | Fwd primer for amplifying N term of PKCII | Sigma | MR52F | CCATCACggatctggcggtagtATGGCTGACCCGGCTGCG |
| Sequence-based reagent | Rvs primer for amplifying N term of PKCII | Sigma | MR52R | GCCCTGAAAATACAGGTTTTCCTTTTCTTCCGGGACCTTGGTTCCC |

*Appendix 1 Continued on next page*

*Appendix 1 Continued*

| Reagent type (species) or resource | Designation | Source or reference | Identifiers | Additional information |
|---|---|---|---|---|
| Sequence-based reagent | Fwd primer for adding stop codon to PKCII | Sigma | MR56F | AGTCAAGAGCTAAgcgg ccgctcgagtctagagcctgc |
| Sequence-based reagent | Rvs primer for adding stop codon to PKCII | Sigma | MR56R | gactcgagcggccgcTTAGCTCTTGA CTTCGGGTTTTAAAAATTCAG |
| Commercial assay or kit | Transcreener ADP2 FI Assay (1000 Assay, 384 Well) | BellBrook Labs | 3013-1K | |
| Chemical compound, drug | Deuterium oxide 99.9% | Sigma | 151882 | |
| Chemical compound, drug | Guanosine 5'-diphosphate (GDP) sodium salt hydrate | Sigma | G7127-100MG | |
| Chemical compound, drug | Guanosine 5'-triphosphate (GTP) sodium salt hydrate | Sigma | G8877-250MG | |
| Chemical compound, drug | Sodium deoxycholate | Sigma | D6750 | |
| Chemical compound, drug | Polyoxyethylene (10) lauryl ether | Sigma | P9769 | |
| Chemical compound, drug | CHAPS, Molecular Biology Grade | EMD Millipore | 220201 | |
| Chemical compound, drug | Phosphatidylserine (Porcine Brain) | Avanti | 840032C | |
| Chemical compound, drug | Phosphatidylethanolamine (Egg yolk) | Sigma | P6386 | |
| Chemical compound, drug | Cholesterol | Sigma | 47,127U | |
| Chemical compound, drug | Phosphatidylcholine (Egg yolk) | Avanti | 840051C | |
| Chemical compound, drug | Phosphatidylinositol-4,5-bisphosphate (Porcine Brain) | Avanti | 840046 | |
| Chemical compound, drug | Sphingomyelin (Egg yolk) | Sigma | S0756 | |
| Chemical compound, drug | 1,2-Dioleoyl-*sn*-glycero-3-phosphocholine (DOPC) | Avanti | 850375C | |
| Chemical compound, drug | 1,2-Dioleoyl-*sn*-glycero-3-phospho-L-serine (18:1, DOPS) | Avanti | 840035C | |
| Chemical compound, drug | 1,2-Dioleoyl-*sn*-glycero-3-phosphoethanolamine-*N*-[4-(p-maleimidomethyl)cyclohexane-carboxamide] (18:1 MCC-PE) | Avanti | 780201C | |
| Chemical compound, drug | 10 mg/mL beta casein solution | Thermo Fisher | 37528 | |
| Chemical compound, drug | 10× PBS (pH 7.4) | Corning | 46-013CM | |
| Chemical compound, drug | glucose oxidase from *Aspergillus niger* (225 U/mg) | Biophoretics | B01357.02 | |
| Chemical compound, drug | Catalase | Sigma | C40-100MG Bovine Liver | |
| Chemical compound, drug | Trolox | Cayman Chemicals | 10011659 | |
| Chemical compound, drug | Dyomics 647 maleimide dye | Dyomics | 647P1-03 | |
| Chemical compound, drug | Coenzyme A | Sigma | C3019 | |

*Appendix 1 Continued on next page*

*Appendix 1 Continued*

| Reagent type (species) or resource | Designation | Source or reference | Identifiers | Additional information |
|---|---|---|---|---|
| Chemical compound, drug | Sulfuric acid | Sigma | 58105-2.5L-PC | |
| Software, algorithm | COOT-0.9.4.1 | CCP4 | https://www2.mrc-lmb.cam.ac.uk/personal/pemsley/coot/ | |
| Software, algorithm | Phenix-1.19.1 | Open source | https://www.phenix-online.org/ | |
| Software, algorithm | PDBePISA (Proteins, Interfaces, Structures and Assemblies) | EMBL-EBI | https://www.ebi.ac.uk/pdbe/pisa/pistart.html | |
| Software, algorithm | ESPript 3.0 | *Robert and Gouet, 2014* | https://espript.ibcp.fr | |
| Software, algorithm | HDExaminer | Sierra Analytics | http://massspec.com/hdexaminer | |
| Software, algorithm | GraphPad Prism 7 | GraphPad | https://www.graphpad.com | |
| Software, algorithm | PyMOL | Schroedinger | http://pymol.org | |
| Software, algorithm | Compass Data Analysis | Bruker | https://www.bruker.com | |
| Software, algorithm | ChimeraX | UCSF | https://www.rbvi.ucsf.edu/chimerax/ | |
| Software, algorithm | ImageJ/Fiji | ImageJ | https://imagej.net/software/fiji/ | |
| Software, algorithm | Nikon NIS elements | Nikon | https://www.microscope.healthcare.nikon.com/products/software/nis-elements | |
| Software, algorithm | cryoSPARC v.3.3.2 | Structura Biotechnology | https://cryosparc.com/ | |
| Other | Sf9 insect cells for expression | Expression Systems | 94–001S | Sf9 cell line used for protein expression (Methods) |
| Other | Insect cell media | Expression Systems | 96-001-01 | Sf9 cell media (Methods) |
| Other | Hellmanex III cleaning solution | Fisher | 14-385-864 | Cleaning solution for TIRF experiments (PMID:36842083) |
| Other | Six-well sticky-side chamber | IBIDI | 80608 | Side chamber used for TIRF experiments (Methods) |
| Other | C-Flat 2/2T grids | Electron Microscopy Sciences | CFT-223C | Grids used for EM studies (Methods) |
| Other | PDB coordinate file for p110γ-NB7 structure | PDB | 8DP0 | PDB to use for p110-NB7 structure |
| Other | EM density file for p110γ-NB7 complex | EMD | EMD-27627 | EM density file to use for p110-NB7 |
| Other | HDX-MS and phosphorylation proteomics data | PRIDE | PXD040765 | Database and number where HDX data was uploaded |

