## [Editor Report · eLife assessment]

This study presents **fundamental** new insight into the regulatory apparatus of PI3Kγ, a kinase in signaling pathways that control the immune response and cancer. A suite of biophysical and biochemical approaches provide **convincing** evidence for new sites of allosteric control over enzyme activity. The rigorous findings provide structure and dynamic information that may be exploited in efforts to control PI3Kγ activity in a therapeutic setting.

---

## [Referee Report · Reviewer #1 (Public Review)]

The authors examine signaling factors that differentiate parallel routes to activating phosphoinositide 3-kinase gamma (PI3Kγ). Dissecting the convergent pathways that control PI3Kγ activity is critical because PI3Kγ is a therapeutic target for treating inflammatory disease and cancer. Here, the authors employ a multipronged approach to reveal new aspects for how p84 and p101 pair with p110γ to activate the PI3Kγ heterodimer. The key instigator to this study is a previously reported inhibitory Nanobody, NB7. The hypothesized mechanism for NB7 allosteric inhibition of p84- p110γ was previously proposed to involve blockage of the Ras-binding domain. The authors revise the allosteric inhibition model based on meticulous profiling of various PI3Kγ complex interactions with NB7. In parallel, a cryo-EM-derived model of NB7 bound to the p110γ subunit convincingly reveals a Nanobody interaction pocket involving the helical domain and regulatory motifs of the kinase domain. This revelation shifts the focus to the helical domain, a known target of PKC phosphorylation. While the connections between NB7 interactions and the effects of PKC phosphorylation are sometimes tenuous, it could be argued that the Nanobody served as a tool to reveal the importance of the helical domain to p110γ regulation.

The sites of PKC-mediated p110γ helical domain phosphorylation were unexpectedly inaccessible in the available structural models. Nevertheless, mass spectrometry (MS)-based phosphorylation profiling indicates that PKC can phosphorylate the helical domain of p110γ and p84/p110γ (but not p101/p110γ) in vitro. The authors hypothesize that helical domain dynamics dictate susceptibility to PKC phosphorylation. To explore this notion, carefully executed, rigorous H/D exchange MS (HDX-MS) experiments were performed comparing phosphorylated vs. unphosphorylated p110γ. Notably, this design reveals more about the consequences of p110γ phosphorylation, rather than the mechanisms of p84/p101 promoting/resisting phosphorylation. Nevertheless, HDX-MS is very well suited to exploring secondary structure dynamics, and helical domain phosphorylation strikingly increases dynamics consistent with increased regional accessibility. The increased dynamics also nicely map to the pocket enveloped by the inhibitory NB7 Nanobody.

Ultimately, this study reveals an unexpected p110γ pocket that allows an engineered Nanobody to allosterically inhibit PI3Kγ complexes. The cryo-EM characterization of the interaction inspired an HDX-MS investigation of known sites of phosphorylation in the region. These insights could be linked to differences/convergences of p84 and p101 complex formation and activation of PI3Kγ, and future work may clarify these mechanisms further. The data presented herein will also be useful for broadening the target surface for future therapeutic developments. New allosteric connections between effector binding sites and post-translational modifications are always welcome.

---

## [Referee Report · Reviewer #2 (Public Review)]

Harris et al. have described the cryo-EM structure of PI3K p110gamma in a complex with a nanobody that inhibits the enzyme. This provided the first structure of full-length of PI3Kgamma in the absence of a regulatory subunit. This nanobody is a potent allosteric inhibitor of the enzyme, and might provide a starting point for developing allosteric, isotype-specific inhibitors of the enzyme. One distinct effect of the nanobody is to greatly decrease the dynamics of the enzyme as shown by HDX-MS, which is consistent with a growing body of observations suggesting that for the whole PI3K superfamily, enzyme activators increase enzyme dynamics.

The most remarkable outcome of the study is that upon observing the site of nanobody binding, the authors searched the literature and found that there was a previous report of a PKCbeta phosphorylation of PI3Kgamma in the helical domain that is near the nanobody binding site. This led the authors to re-examine the consequence of the phosphorylation armed with better structural models and the tools to study the effects of this phosphorylation on enzyme dynamics. They found that the site of phosphorylation is buried in the helical domain, suggesting that a large conformational change would have to take place to enable the phosphorylation. HDX-MS showed that phosphorylation at three sites clustered in the helical domain generate a distinctly different conformation with rapid deuterium exchange. This suggests that the phosphorylation locks the enzyme in a more dynamic state. Their enzyme kinetics show that the phosphorylated, dynamic enzyme is activated.

While this phosphorylation was reported before, the authors have provided a mechanism for why this activates the enzyme, and they have shown why binders that stabilise the helical domain (such as binding to the p101 regulatory subunit and the nanobody) prevent the phosphorylation. It is this insight into the dynamics of the PI3Kgamma that will likely be the long-lasting influence of the work.

The paper is well written and the methods are clear.

---

## [Author Response]

The following is the authors' response to the original reviews.

We appreciate the in depth review of our manuscript, and the excellent suggestions from the two reviewers. We have addressed all concerns as described in the point by point response below. We have also added all of these changes to a revised version posted to biorxiv on May 23rd 2023 (BIORXIV/2023/536585).

**Reviewer #1 (Recommendations For The Authors):**
It is sometimes difficult to connect the rationalizations behind the transitions between NB7 binding interaction, the compare/contrast of p84 and p101 effectors, and the synergy with phosphorylation. More explanation of the rationalizations behind these transitions in the Results would be helpful.

We agree that the manuscript would benefit from better transitions between the sections. We have added a new paragraph in the final section describing the nanobody structure before the helical domain phosphorylation that fully describes the rationale for how both inform on the critical role of helical domain dynamics in kinase activity. This paragraph is shown below.

_‘The interface of NB7 with p110_g _is distant from both the putative membrane binding surface, as well as the catalytic machinery of the kinase domain. To further understand how this nanobody could so potently inhibit PI3K activity we examined any other potential modulators of PI3K activity localised in this region. There are two regulatory phosphorylation sites in the helical (Walser et al., 2013) and kinase domain (Perino et al., 2011) localised at the NB7 interface. This is intriguing as helical domain phosphorylation is activating, and kinase domain phosphorylation is inhibitory. This suggested a critical role in the regulation of p110_g *is the dynamics of this kinase-helical interface. To fully define the role of NB7 in altering the dynamics of the helical domain we needed to study other modulators of helical domain dynamics.’*

The Methods section would benefit from careful copy editing for clarity and consistency.

We have gone through the methods section and edited for clarity and consistency throughout.

There's a minor ambiguity throughout when referring to the phosphorylation of S594/S595. Although close inspection makes it clear that this refers to the monophosphorylation of either site S594 or S595, there are several references to "S594/S595" that could be interpreted as phosphorylation of both residues.

We agree that this was ambiguous in the original text. We have added an explicit statement describing this as a single phosphorylation event.

*‘The modification at this site results in a single phosphorylation event , but due to CID MS/MS fragmentation we cannot determine which site is modified, and will be described as S594/S595 throughout the manuscript.’*

In Figure 2B, the authors show the cryo-EM density map and the structural model based on this map. It would be helpful to also include an image of the structural model fit in the density map to allow readers to evaluate the quality of the map and model. The 2F panel provides an important view of this fit, but CD3 models are difficult to discern.

We agree that this would help interpret model quality. We have added a new supplemental figure showing the fit of both p110 and NB7 into this Cryo EM density (see new Fig S2).

Paragraph starting at line 258: The shift to monitoring ATPase activity is confusing here. ATPase activity indicates production of ADP + phosphate (rather than ADP + PIP3). However, an explanation is provided that states that measuring ADP production serves as a surrogate for measuring PIP3 production. The apparent absence of membrane PIP2 substrate in Figure 4E (left) suggests that there is a true ATPase background activity in this kinase. If so, does the increase in ADP production in Fig. 4E reflect the inclusion of PIP2 substrate, increased background ATPase activity, or both?

We agree that this was worded confusingly in the original version. We have now clarified exactly what we are observing in these ATPase assays. The new paragraph is appended below

*‘To further explore the potential role of phosphorylation in mediating p110g activity, we examined the kinase activity of p110g under two conditions: basal ATP turnover, and with PIP2 containing lipid membranes. The experiments in the absence of PIP2 measure turnover of ATP into ADP and phosphate, and is a readout of basal catalytic competency. Experiments with PIP2, measured ATP consumed in the generation of PIP3, as well as in non-productive ATP turnover. The p110g enzyme in the absence of stimulators is very weakly active towards PIP2 substrate with only ~2 fold increased ATP turnover compared to in the absence of membranes. This is consistent with very weak membrane recruitment of p110g complexes in the absence of lipid activators (Rathinaswamy et al., 2023). PKCb-mediated phosphorylation enhanced the ATPase activity of p110g ~2-fold in both the absence and presence of membranes (Fig. 4E). This suggests that the effect of phosphorylation is to change the intrinsic catalytic efficiency of phosphorylated p110g, with limited effect on membrane binding.’*

In the section "Nanobody blocks p110gamma phosphorylation," it's not entirely convincing that "the presence of NB7 showed even lower phosphorylation than p110gamma-p101." This does not seem to be subject to a significance test in Figure 5A/B. The follow-up point about "complete abrogation of phosphorylation," however, is readily apparent.

We agree that we could have been more precise with our language, as this is not a complete block of phosphorylation, it is merely a significant decrease in phosphorylation. We have removed the comparison to p110-p101, and also removed the statement about complete abrogation of phosphorylation. This is now reworded to

*‘The presence of NB7 showed a significant decrease in p110g phosphorylation at both sites (Fig. 5A-B).’*

Figure 1: Legend needs to include more detail to define the data. (1A) Representations of variance need to be clarified (e.g., replicates, error bar meaning). Consider "Normalized lipid kinase activity" as a y-axis label and expand on the activity measurement and normalization in the legend. (1B) How was error calculated? (1C) Mislabeled as 1B? Also, consider clarifying the first title highlighting the comparison to class IA PI3Ks. (1D) Typo: "Y647-p84/p110gamma." Also, would it not be more accurate to say "effect of nanobody NB7 on PI3K displacement..." for this experiment?

We apologise for these oversights. See details on what has been changed in Fig 1A, 1B, 1C, and 1D.

For Fig 1A we now show the data where each replicate is indicated in the graph in the absence of error bars, and have also more clearly expanded on this activity measurement in the figure legend and also stated the replicate number.

For Fig 1B we now clearly state how the error was generated.

For Fig 1C we have fixed the typo

For Fig 1D, we have fixed this typo and also changed the sub-heading as suggested.

New figure legend is below as well

**
*Figure 1. The inhibitory nanobody NB7 binds tightly to all p110γ complexes and inhibits kinase activity, but does not prevent membrane binding*
**

**
*A.*
**
*Cartoon schematic depicting nanobody inhibition of activation by lipidated Gβγ (1.5 µM final concentration). Lipid kinase assays show a potent inhibition of lipid kinase activity with increasing concentrations of NB7 (3-3000 nM) for the different complexes. Experiments are carried out in triplicate (n=3) with each replicate shown. The y-axis shows lipid kinase activity normalised for each complex activated by Gβγ in the absence of nanobody. Concentrations of each protein were selected to give a lipid kinase value in the detectable range of the ATPase transcreener assay. The protein concentration of p110γ (300 nM), p110γ-p84 (330 nM) and p110γ-p101 (12 nM) was different due to intrinsic differences of each complex to be activated by lipidated Gβγ, and is likely mainly dependent for the difference seen in NB7 response.*

**
*B*
**
*. Association and dissociation curves for the dose response of His-NB7 binding to p110γ, p110γ-p84 and p110γ-p101 (50 – 1.9 nM) is shown. A cartoon schematic of BLI analysis of the binding of immobilized His-NB7 to p110γ is shown on the left. Dissociation constants (KD) were calculated based on a global fit to a 1:1 model for the top three concentrations and averaged with error shown. Error was calculated from the association and dissociation value (n=3) with standard deviation shown. Full details are present in the source data.*

**
*C*
**
*. Association and dissociation curves for His-NB7 binding to p110γ, p110a-p85a, p110b-p85b, and p110d-p85b. Experiments were performed in duplicate with a final concentration of 50 nM of each class I PI3K complex.*

**
*D*
**
*. Effect of NB7 on PI3K recruitment to supported lipid bilayers containing H-Ras(GTP) and farnesyl-Gbg as measured by Total Internal Reflection Fluorescence Microscopy (TIRF-M). DY647-p84/p110g displays rapid equilibration kinetics and is insensitive to the addition of 500 nM nanobody (black arrow, 250 sec) on supported lipid bilayers containing H-Ras(GTP) and farnesyl-Gbg.*

**
*E*
**
*. Kinetics of 50 nM DY647-p84/p110g membrane recruitment appears indistinguishable in the absence and presence of nanobody. Prior to sample injection, DY647-p84/p110g was incubated for 10 minutes with 500 nM nanobody.*

**
*F*
**
*. Representative TIRF-M images showing the localization of 50 nM DY647-p84/p110g visualized in the absence or presence of 500 nM nanobody (+NB7). Membrane composition for panels C-E: 93% DOPC, 5% DOPS, 2% MCC-PE, Ras(GTP) covalently attached to MCC-PE, and 200 nM farnesyl-Gbg.*

Figure 2: (1A) For consistency with the rest of the paper, p110g can be updated with the Greek character. (1B) This may have been intentional to attract attention to subdomains interacting with NB7, but "colored according to the schematic" omits the purple RBD. (2F) the figure legend should specify whether p110gamma surfaces depicted are the cryo-EM density or a surface rendition of the structural model.

We agree and have fixed the p110 typo, and have also colored the schematic the same as shown in the cartoon model.

The data shown in Fig 1B is indeed the Cryo EM density and this is now clearly indicated in the legend.

Figure 3: (3B) Specifying the [M+H] as [M+2H]2+ and [M+4H]4+ would help the reader understand the delta mass for monophosphorylation here. Given the broad readership of this journal, it would be useful to define 't' and 'e' as 'theoretical' and 'experimental' in the legend. It may also help to be explicit about the meaning of the red spectra and residues in the legend. (3C-E) autocorrect typo for "(C)" and an opportunity to update "b" for Greek character beta.

We agree that clearly defining the charge state of each spectra will make it more obvious that we are dealing with a mono-phosphorylation and have made this change as suggested in the figure. We have also clearly define m/z t and m/z e in the figure legend, as well as the black and red lines, and characters. Finally we have added PKCb for all descriptions of PKC treatment in the figure, and fixed the incorrect PKC’b’ in the legend.

Figure 4: (4C) Given the common use of "ND" for other terms, it would be useful to spell out "no deuterium" or "undeuterated." (4E) the parenthetical "(concentration, 12nM to 1000nM)" could be clarified. How are the (presumably p100gamma) concentration ranges reflected in the three plotted data points per treatment? See also Figure 5E.

We agree and have redefined ND as undeuterated. We apologise for the typo in the figure legend, as the concentrations of p110 gamma were the same for both phosphorylated and non-phosphorylated, with this being a typo (all concentrations of enzyme were 1000 nM). This has been changed here and in Fig 5E.

Figure 5: (5A/B) Some clarification that we're looking at extracted ion chromatograms would be very useful in this figure legend. On a related note, the experimental details on the LC-MS methodology for this data appear to be split between two sections of the Methods: the "Phosphorylation analysis" paragraph (line 526) and the HDX-MS section. Some more explicit cross-referencing would clarify this experiment. (5E) Clarify inclusion of PIP2 membranes here.

We have clearly described that we are looking at extracted ion chromatograms in both panel A and B. We also have normalized the experimental methods in the LC-MS as these used exactly the same procedure. Finally, we now clearly describe the assays shown in Fig 5E were performed in the absence of PIP2 membranes.

Miscellaneous typos:Line 205: reference omitted for "Previous study.."

We have added this reference

Line 196: "unambiguous"

Fixed to unambiguously

**Reviewer #2 (Recommendations For The Authors):**
The only mistake I spotted was that on line 729 there is a reference to Fig 3C that should actually be Fig. 4C

We have changed this to the correct Fig 4C.